# Lifelong Neural Predictive Coding: Learning Cumulatively Online without Forgetting

**Alexander G. Ororbia**
Rochester Institute of Technology
Rochester, NY 14623, USA
ago@cs.rit.edu

**Ankur Mali**
University of South Florida
Tampa, FL 33620, USA
ankurarjunmali@usf.edu

**C. Lee Giles**
The Pennsylvania State University
State College, PA 16801, USA
clg20@psu.edu

**Daniel Kifer**
The Pennsylvania State University
State College, PA 16801, USA
duk17@psu.edu

## Abstract

In lifelong learning systems based on artificial neural networks, one of the biggest obstacles is the inability to retain old knowledge as new information is encountered. This phenomenon is known as catastrophic forgetting. In this paper, we propose a new kind of connectionist architecture, the Sequential Neural Coding Network, that is robust to forgetting when learning from streams of data points and, unlike networks of today, does not learn via the popular back-propagation of errors. Grounded in the neurocognitive theory of predictive coding, our model adapts its synapses in a biologically-plausible fashion while another neural system learns to direct and control this cortex-like structure, mimicking some of the task-executive control functionality of the basal ganglia. In our experiments, we demonstrate that our self-organizing system experiences significantly less forgetting compared to standard neural models, outperforming a swath of previously proposed methods, including rehearsal/data buffer-based methods, on both standard (SplitMNIST, Split Fashion MNIST, etc.) and custom benchmarks even though it is trained in a stream-like fashion. Our work offers evidence that emulating mechanisms in real neuronal systems, e.g., local learning, lateral competition, can yield new directions and possibilities for tackling the grand challenge of lifelong machine learning.

## 1 Introduction

Lifelong learning is a part of machine learning and artificial intelligence research with the goal of developing a computational agent that can learn over time and continually accumulate new knowledge while retaining its previously learned experiences [1, 2]. For example, suppose an agent needs to learn to classify digits, then types of clothing, and then sketches of objects. As each new task arrives the agent is expected to process the accompanying data and learn the new task but still remember how to complete the old tasks, at least without significant degradation in performance or loss of generalization. Modern-day connectionist systems are typically trained on a fixed pool of data samples, collected in controlled environments, in isolation and random order, and then evaluated on a separate validation data pool. This is a far cry from what we really desire from learning machines.

When we look to humans or other animals, we see that they are more than capable of learning in a continual manner, making decisions based on sensorimotor input throughout their lifespans [3]. This ability to incrementally acquire and refine knowledge over long periods of time is driven by cognitive processes that come together to create the experience-driven specialization of motor and perceptual

36th Conference on Neural Information Processing Systems (NeurIPS 2022).

skills [4, 3]. Thus, evaluating how neural systems generalize on task sequences, as opposed to single, isolated tasks, proves to be a far greater challenge. In order to continually adapt, the brain must retain specific memories of prior tasks. In working towards the challenge of lifelong machine learning, this paper makes the following contributions: 1) we propose the sequential neural coding network, an interactive generative model that jointly reconstructs input and predicts its label, and an algorithm for updating its synapses, 2) we show that memory retention is vastly improved by integrating our model's multi-step nature with lateral competition that is driven by a task selection function inspired by the basal ganglia [5], and 3) we compare our model's performance against state-of-the-art baselines, including both regularization and rehearsal/replay-based methods, on publicly-available benchmarks.

## 2 Related Work

It is well-known that when artificial neural networks (ANNs) are trained on more than one task sequentially, the new information contained in subsequent tasks leads to catastrophic interference (a.k.a. catastrophic forgetting) with the information acquired in earlier tasks [6, 7, 8]. This happens in connectionist systems when the new data instances to be learned are significantly different from previously observed ones. This causes the new information to overwrite knowledge currently encoded in the system's synaptic weights, due to the sharing of neural representations over tasks [9, 6] (this is known as the representational overlap problem). In isolated, task-specific (offline) learning (though it still occurs [10]), this type of weight overwriting occurs to a lesser degree because the patterns are presented to the agent pseudo-randomly and multiple times, i.e., via multiple epochs.

Over the decades, there have been many approaches proposed to mitigate catastrophic forgetting in neural systems. Some of the earliest attempts proposed memory systems where prior data points were stored and regularly replayed to the network in a process called "rehearsal", which involved interleaving these data points with samples drawn from new datasets [11, 12, 13, 14, 15]. Though effective, the main drawback of these approaches is that they require explicitly storing old data. Such a mechanism is not known to exist in the brain and, as a practical matter, this leads to exploding working (hardware) memory requirements (inefficiency). In addition, rehearsal-based approaches do not offer any mechanisms to preserve consolidated knowledge in the face of acquiring new information [4]. Other approaches attempt to allocate additional neural "resources", i.e., growing the networks when required [16, 17], motivated by earlier findings [18]. However, this leads to dramatically increasing computational requirements over time as the networks grow larger. To compound these issues further, systems with growing capacity cannot know how many resources to allocate at a given time since the number of tasks and samples are not known a priori (without imposing strong assumptions on the input distribution). Other approaches try to block old information from being overwritten through regularization [19]. From this vast collection of research, each approach bearing strengths and weaknesses, three suggested remedies have emerged: 1) allocate additional neural resources to accommodate new knowledge, 2) use non-overlapping representations (or semi-distributed ones [20]) if resources are fixed, and 3) interleave old patterns with new ones as new information is acquired.

In this work, we consider the setting where the space available to the agent grows slowly compared to the rate of new tasks being presented. This means that we cannot just create a new, separate network for every task observed in the stream. Furthermore, storing, reshuffling, and re-presenting data in the stream is not feasible in this setting. Concretely, our approach could be classified as class incremental learning (Class-IL) [21] given that we **do not use task identifiers** at both training and test time. In addition to addressing the above, our contribution to lifelong learning is motivated by the premise that developing algorithms [22, 23, 24, 25, 26] that serve as alternatives to back-propagation will lead to the creation of promising architectures with mechanisms equipped to tackle catastrophic interference.

## 3 Cumulative Learning with Neural Coding

**Notation:** We start by defining key notation. $\odot$ indicates a Hadamard product while $\cdot$ denotes matrix/vector multiplication. $(\mathbf{v})^T$ is the transpose of $\mathbf{v}$. Matrices/vectors are depicted in bold-font, e.g., matrix $\mathbf{M}$ or vector $\mathbf{v}$ (scalars shown in italic). Finally, $||\mathbf{v}||_2$ denotes the Euclidean norm of $\mathbf{v}$.

### 3.1 Sequential Learning and the Data Continuum

This work focuses on adapting a neural system in the context of sequential learning. Starting from an early definition [27] of this form of learning, we assume that there is a sequence of tasks $\mathcal{T}_1, \mathcal{T}_2, \mathcal{T}_3, \dots$ (with potentially no end) with each task presented to a system in order. When faced with the $(N+1)$th task, the system should use the knowledge that it has gained from the previous $N$ tasks to aid in learning and performing the current task. The knowledge of a system is stored in a knowledge base (KB), e.g., the synapses of a neural model. Each task $\mathcal{T}_i$ has its own corresponding dataset $D_i = \{(\mathbf{y}_1, \mathbf{x}_1, t_i) \dots (\mathbf{y}_{n_i}, \mathbf{x}_{n_i}, t_i)\}$ with $n_i$ examples. Here $\mathbf{x}_j \in \mathcal{R}^{J_x \times 1}$ represents the feature vector of the $j^{\text{th}}$ example ($J_x$ is its dimensionality), $\mathbf{y}_j \in \{0, 1\}^{J_y \times 1}$ is the target (label), and $t_i$ is the task descriptor that identifies $(\mathbf{y}_j, \mathbf{x}_j)$ as being a data point from task $i$. The task descriptor is one-hot encoded as $\mathbf{t} \in \{0, 1\}^{(N+1) \times 1}$ and, when a new task is encountered, the size of the one-hot vector is increased by one – thus the network adds an extra randomly initialized input node if it accepts $t_i$ as an extra input. Note that while we present the task data continuum with $t_i$ explicitly depicted, we will generalize our models to not depend on $t_i$ (making this problem task descriptor-free).

**Dynamic Output Units:** The output nodes in our setting get re-used for each new task. For example, output node 1 of the network could represent a prediction for the digit "1" in task $\mathcal{T}_1$ (e.g., digit recognition), while in task $\mathcal{T}_2$ (e.g., clothing recognition) the same node could represent a prediction for "pants". If a new task has more classes than previous tasks, we add output nodes with randomly initialized weights. For example, if prior tasks were binary and the new task has $4$ targets, we add $2$ more outputs to the network. Note this is a difficult form of cumulative learning [28].

**Context Units:** For the model that we develop in this study, when task $t$ is encountered, for each layer $\ell$ in the network, we make use of a task embedding vector $\mathbf{g}_t^\ell$ (this new memory is much smaller than creating a new network for task $t$, which would require new weight matrices per layer rather than an extra vector). All $\mathbf{g}_t^\ell \in \mathcal{R}^{J_\ell \times 1}$ ($J_\ell$ is the number of units in layer $\ell$) are stored in memory matrices $\mathbf{M}^\ell \in \mathcal{R}^{(N+1) \times J_\ell}$ ($\mathcal{M} = \{\mathbf{M}^1, \dots \mathbf{M}^L\}$), where a context can be retrieved using a one-hot encoding of the task descriptor, i.e., $\mathbf{M}^\ell \cdot \mathbf{t}$ ( $\mathbf{t}$ will be produced by another system – see Section 3.3).

### 3.2 The Interactive Generative Model

The sequential neural coding network (S-NCN) is designed to make flexible conditional predictions – e.g., predicting $\mathbf{y}$ given $\mathbf{x}$, predicting both $\mathbf{y}$ and $\mathbf{x}$, predicting missing parts of $\mathbf{x}$ given $\mathbf{y}$ and the observed parts of $\mathbf{x}$, etc. In order to do so, it treats inputs/outputs in a non-standard way. The input to the model is the set of task contexts $\{\mathbf{g}_t^1, \dots, \mathbf{g}_t^L\}$ and the output units represent $(\mathbf{y}, \mathbf{x})$. To predict $\mathbf{y}_i$ given $\mathbf{x}_i$, we clamp the output nodes responsible for predicting $\mathbf{x}$, forcing their output to be $\mathbf{x}_i$.[1] During training, outputs are clamped to both $\mathbf{y}_i$ and $\mathbf{x}_i$, forcing the S-NCN to update latent states and synapses. The S-NCN can also operate as a probabilistic generative network by feeding in a random noise vector as input, but we leave this extension to future work (and focus on predicting $\mathbf{y}$ given $\mathbf{x}$).

The full computational process of the S-NCN is defined by three key steps: 1) layer-wise hypothesis generation, 2) state error-correction, and 3) synaptic weight evolution. The first two steps iteratively predict and correct the representations of the model for the input and target values of the task. After $K$ iterations, model weights and the current task context memory are updated. In this section, we provide details of the above steps and then describe the objective function that our model optimizes.

#### 3.2.1 Inference: Predicting and Correcting States

**Layer-wise State Prediction.** The architecture of the S-NCN can be viewed as a stack of parallel, stateful neural-based prediction layers $P_1, \dots, P_\ell, \dots, P_m$, where the goal of each predictor is to guess the internal state of the predictor in the layer below (i.e., the S-NCN is not a feedforward network). The state of $P_\ell$ (layer $\ell$) is represented by the (zero-initialized) vector $\mathbf{z}^\ell \in \mathcal{R}^{J_\ell \times 1}$ ($J_\ell$ is the number of units in layer $\ell$). $P_\ell$ makes a prediction $\mathbf{z}_\mu^{\ell-1}$ about the current state $\mathbf{z}^{\ell-1}$ of $P_{\ell-1}$ (layer $\ell - 1$). Furthermore, we let $\mathbf{z}_x^0$ and $\mathbf{z}_y^0$ denote clamped outputs. That is, if we want to predict the label $\mathbf{y}_i$ given the features $\mathbf{x}_i$, we set $\mathbf{z}_x^0 = \mathbf{x}_i$ and if we want to train, we set both $\mathbf{z}_x^0 = \mathbf{x}_i$ and $\mathbf{z}_y^0 = \mathbf{y}_i$. The

---

[1]Similarly, in the case of missing data, we can ask the network to predict the $\mathbf{y}_i$ and the missing parts of $\mathbf{x}_i$ given the observed parts of $\mathbf{x}_i$ by clamping outputs to only the observed parts of $\mathbf{x}_i$. See appendix for details.

values predicted by layer 1 are denoted by $\mathbf{z}^0_{\mu,x}$ and $\mathbf{z}^0_{\mu,y}$. Note that $\mathbf{z}^0 = [\mathbf{z}^0_x, \mathbf{z}^0_y]$, where the two are concatenated (if either is missing, the SNCN completes the required values – see Appendix).

With respect to neural structure, the parallel predictors of the S-NCN are locally connected through (forward) generative weights $\mathbf{W}^\ell \in \mathcal{R}^{J_\ell \times J_{\ell+1}}$ and error (feedback) weights $\mathbf{E}^\ell \in \mathcal{R}^{J_{\ell+1} \times J_\ell}$, which work to transmit error information to relevant regions of neural processing elements, effectively coordinating all of the S-NCN's predictors. Formally, a predictor (which embodies the computation of deep excitatory pyramidal neurons), with state $\mathbf{z}^{\ell+1}$, that guesses the state of $\mathbf{z}^\ell$, takes on the following form (given matrix $\mathbf{W}^{\ell+1}$ and activation $\phi^{\ell+1}$):

$$\mathbf{z}^\ell_\mu = \mathbf{W}^{\ell+1} \cdot \phi^{\ell+1}(\mathbf{z}^{\ell+1}), \quad \mathbf{e}^\ell = (\mathbf{z}^\ell - \mathbf{z}^\ell_\mu) \tag{1}$$

where $\mathbf{e}^\ell$ is a block of error units. Error units are paired with each predictor. Their task is to compute the disagreement or mismatch between the predictor's output and the target activity pattern $\mathbf{z}^\ell$ (embodying the mismatch computation carried out by superficial pyramidal neurons). The error unit vector $\mathbf{e}^\ell$ can also be derived from the total discrepancy reduction presented in Equation 5 (see Appendix). Note that for layer 0, $\mathbf{e}^0 = [\mathbf{e}^0_x, \mathbf{e}^0_y]$ (there are error neurons for $\mathbf{x}_i$ as well as others for $\mathbf{y}_i$, the concatenation of which makes up the bottom-most prediction error signal).

**Latent State Correction and Context Updating.** Once each layer $\ell$ has made a prediction about the layer below it and error units have been activated, the state of layer $\ell$ can be adjusted to take into account the local top-down and bottom-up error information. Using its current state and the error nodes (along with its task context embedding $\mathbf{g}^\ell_t$), layer $\ell$ in the S-NCN adjusts its state $\mathbf{z}^\ell$ as follows:

$$\mathbf{z}^\ell(k) = f^\ell(\mathbf{z}^\ell(k-1) + \beta \mathbf{d}^\ell, \mathbf{g}^\ell_t), \quad \text{where } \mathbf{d}^\ell = -\mathbf{e}^\ell + \mathbf{E}^\ell \cdot \mathbf{e}^{\ell-1} \tag{2}$$

where $\beta$ is the state correction rate and $k$ marks one step of the S-NCN's $K$-step inference process. Note that $\mathbf{d}^\ell$ is the perturbation that adjusts the values of the hidden states – it combines a top-down expectation of layer $\ell$ with a bottom-up pressure from layer $\ell-1$. The error feedback weights $\mathbf{E}^\ell$ are parameters that play a crucial role in this calculation, as they are responsible for transmitting the error at $\ell$ back up to layer $\ell+1$. Notably, part of the state-correction requires competition among the individual units in a given layer through the function $f^\ell(\mathbf{z}^\ell, \mathbf{g}^\ell_t)$. There are various ways in which this function can be implemented and, in this study, we implemented it as follows:

$$f^\ell(\mathbf{z}^\ell, \mathbf{g}^\ell_t) = (\mathbf{I} \odot \mathbf{V}^\ell) \cdot \mathbf{z}^\ell, \text{ where, } \mathbf{V}^\ell = \text{BKWTA}(\mathbf{g}^\ell_t, K) \cdot \text{BKWTA}((\mathbf{g}^\ell_t)^T, K)$$

where $\text{BKWTA}(\mathbf{v}, K)$ is the binarized $K$ winners-take-all function, yielding a binary vector with 1 at the index of each of the $K$ winning units, or formally:

$$\text{BKWTA}(\mathbf{v}, K) = \big\{ 1 \text{ if } v_j \in \{K \text{ largest elements of } \mathbf{v}\} \text{ and } \phi(\mathbf{v}) = 0 \text{ otherwise} \big\}.$$

In the Appendix, we study other forms of the competition function (we found that the above performed best, so we report this in the main paper). Note that this lateral inhibition is a function of evolving context $\mathbf{g}^\ell$, triggered by a task pointer $t_i$ (produced by a task selector model, which we define later).

In real neural systems, intra-layer competition between units is thought to facilitate contextual processing [29], where only some neuronal signals are strengthened while the activity of others is suppressed. Moreover, lateral competition, often classically modeled with anti-Hebbian learning [30], encourages the formation of sparse codes [31, 32]. Since the S-NCN is an interactive model, incorporating lateral competition is natural and computed online, highlighting its flexibility compared to ANNs.[2] Crucially, this lateral activity is an inductive bias that leads the S-NCN to acquire task-dependent representations that encode information for multiple, disjoint tasks. In a sense, this task specialization that we build into the neural dynamics is similar in spirit to activation sharpening [33].

### 3.2.2 Updating Synaptic Parameters

Given that the S-NCN is an interactive network [34], inferring its states requires running Equations 1 and 2 $K$ times, where the model alternates between making predictions and then correcting states once error units have been computed. Once latent states have been inferred, the S-NCN then adjusts its synaptic values. The synaptic updates take the form of Hebbian-like rules:

$$\Delta \mathbf{W}^\ell = \mathbf{e}^\ell \cdot (\phi^\ell(\mathbf{z}^{\ell+1}))^T \odot \mathbf{S}^\ell_W, \text{ and, } \Delta \mathbf{E}^\ell = \alpha \Big( \mathbf{d}^{\ell+1} \cdot (\mathbf{e}^\ell)^T \Big) \odot \mathbf{S}^\ell_E \tag{3}$$

---

[2]Recurrent weights could model lateral activity in ANNs but this would require using backprop through time.

| **Algorithm 1** State inference procedure. | **Algorithm 2** Weight update computation. |
|---|---|
| 1: **Input:** sample $(\mathbf{y},\mathbf{x},\mathbf{t})$, $\beta$, $\mathcal{M}$, & $\Theta$ | 1: **Input:** $\Lambda$, $\lambda$, $\gamma$, $\mathcal{M}$, & $\Theta$ |
| 2: **function** INFERSTATES$((\mathbf{y},\mathbf{x},\mathbf{t}),\Theta)$ | 2: **function** UPDATEWEIGHTS$(\Lambda,\Theta)$ |
| 3: $\quad (\mathbf{g}^1,...,\mathbf{g}^\ell,...,\mathbf{g}^L) \leftarrow getContexts(\mathbf{t},\mathcal{M})$ | 3: $\quad$ // Calculate weight displacements |
| 4: $\quad$ Set $\mathbf{z}^1,...,\mathbf{z}^\ell,...,\mathbf{z}^L$ to $\mathbf{0}$ | 4: $\quad$ **for** $\ell = L$ to $2$ **do** |
| 5: $\quad \mathbf{e}^L = \mathbf{0}$, $\mathbf{z}_x^0 = \mathbf{x}$, $\mathbf{z}_y^0 = \mathbf{y}$ | 5: $\quad\quad \Delta\mathbf{W}^\ell = (\mathbf{e}^{\ell-1}\cdot(\phi^\ell(\mathbf{z}^\ell))^T)\odot\mathbf{S}_W^\ell$ |
| 6: $\quad$ **for** $k=1$ to $K$ **do** | 6: $\quad\quad \Delta\mathbf{E}^\ell = \gamma(\mathbf{d}^\ell\cdot(\mathbf{e}^{\ell-1})^T)\odot\mathbf{S}_E^\ell$ |
| 7: $\quad\quad$ **for** $\ell = L$ to $1$ **do** $\quad\triangleright$ Run layerwise predictors | 7: $\quad \Delta\mathbf{W}_x^1 = (\mathbf{e}_x^0\cdot(\phi^1(\mathbf{z}^1))^T)\odot\mathbf{S}_{W,x}^1$ |
| 8: $\quad\quad\quad$ **if** $\ell > 1$ **then** $\quad\triangleright$ Latent prediction layer | 8: $\quad \Delta\mathbf{E}_x^1 = \gamma(\mathbf{d}^1\cdot(\mathbf{e}_x^0)^T)\odot\mathbf{S}_{E,x}^1$ |
| 9: $\quad\quad\quad\quad \mathbf{z}_\mu^{\ell-1} = \mathbf{W}^\ell\cdot\phi^\ell(\mathbf{z}^\ell)$ | 9: $\quad \Delta\mathbf{W}_y^1 = (\mathbf{e}_y^0\cdot(\phi^1(\mathbf{z}^1))^T)\odot\mathbf{S}_{W,y}^1$ |
| 10: $\quad\quad\quad\quad \mathbf{e}^{\ell-1} = (\mathbf{z}^{\ell-1}-\mathbf{z}_\mu^{\ell-1})$ | 10: $\quad \Delta\mathbf{E}_y^1 = \gamma(\mathbf{d}^1\cdot(\mathbf{e}_y^0)^T)\odot\mathbf{S}_{E,y}^1$ |
| 11: $\quad\quad\quad$ **else** $\quad\triangleright$ Sensory prediction layer | 11: |
| 12: $\quad\quad\quad\quad \mathbf{z}_{\mu,x}^{\ell-1} = \mathbf{W}_x^\ell\cdot\phi^\ell(\mathbf{z}^\ell)$ | 12: $\quad$ // Update current weights |
| 13: $\quad\quad\quad\quad \mathbf{e}_x^{\ell-1} = (\mathbf{z}_x^{\ell-1}-\mathbf{z}_{\mu,x}^{\ell-1})$ | 13: $\quad$ **for** $\ell = L$ to $2$ **do** |
| 14: $\quad\quad\quad\quad \mathbf{z}_{\mu,y}^{\ell-1} = \mathbf{W}_y^\ell\cdot\phi^\ell(\mathbf{z}^\ell)$ | 14: $\quad\quad \mathbf{W}^\ell = \mathbf{W}^\ell + \lambda\Delta\mathbf{W}^\ell$ |
| 15: $\quad\quad\quad\quad \mathbf{e}_y^{\ell-1} = (\mathbf{z}_y^{\ell-1}-\mathbf{z}_{\mu,y}^{\ell-1})$ | 15: $\quad\quad \mathbf{E}^\ell = \mathbf{E}^\ell + \lambda\Delta\mathbf{E}^\ell$ |
| 16: $\quad\quad$ **for** $\ell = L$ to $1$ **do** $\quad\triangleright$ Correct internal states | 16: $\quad \mathbf{W}_x^1 = \mathbf{W}_x^1 + \lambda\Delta\mathbf{W}_x^1$ |
| 17: $\quad\quad\quad$ **if** $\ell == 1$ **then** | 17: $\quad \mathbf{E}_x^1 = \mathbf{E}_x^1 + \lambda\Delta\mathbf{E}_x^1$ |
| 18: $\quad\quad\quad\quad \mathbf{d}^\ell = -\mathbf{e}^\ell + \mathbf{E}_x^\ell\cdot\mathbf{e}_x^{\ell-1} + \mathbf{E}_y^\ell\cdot\mathbf{e}_y^{\ell-1}$ | 18: $\quad \mathbf{W}_y^1 = \mathbf{W}_y^1 + \lambda\Delta\mathbf{W}_y^1$ |
| 19: $\quad\quad\quad$ **else** | 19: $\quad \mathbf{E}_y^1 = \mathbf{E}_y^1 + \lambda\Delta\mathbf{E}_y^1$ |
| 20: $\quad\quad\quad\quad \mathbf{d}^\ell = -\mathbf{e}^\ell + \mathbf{E}^\ell\cdot\mathbf{e}^{\ell-1}$ | 20: $\quad$ Update $(\mathbf{g}^1,\cdots,\mathbf{g}^L,\mathcal{M})$ via Eqn. 4 |
| 21: $\quad\quad\quad \mathbf{z}^\ell = f^\ell\big(\phi^\ell(\mathbf{z}^\ell) + \beta\mathbf{d}^\ell, \mathbf{g}^\ell\big)$ | 21: $\quad$ // Return new weights |
| 22: $\quad \Lambda = (\mathbf{z}^1,...,\mathbf{z}^\ell,...,\mathbf{z}^L,\mathbf{d}^1,...,\mathbf{d}^\ell,...,\mathbf{d}^L,$ | 22: $\quad \Theta = \{\mathbf{W}_x^1,\mathbf{W}_y^1,...,\mathbf{W}^\ell...\mathbf{W}^L,$ |
| 23: $\quad\quad \mathbf{e}^0,...,\mathbf{e}^\ell,...,\mathbf{e}^{L-1})$ | 23: $\quad\quad \mathbf{E}_x^1,\mathbf{E}_y^1,...,\mathbf{E}^\ell,...,\mathbf{E}^L\}$ |
| 24: $\quad$ **Return** $\Lambda$ | 24: $\quad$ **Return** $\Theta$ |

where $\alpha$ is a scaling factor, usually set to $< 1.0$, that makes the error feedback weights change at a slower rate than the forward weights (this improves convergence [35]). $\mathbf{S}_W^\ell \in \mathcal{R}^{J_\ell \times J_{\ell+1}}$ and $\mathbf{S}_E^\ell \in \mathcal{R}^{J_{\ell+1} \times J_\ell}$ are modulation factors that provide stability to the weight updates (see Appendix).

An important property of the above weight update rules is that they are local – to compute changes in the synapses, all we require is the information immediately available to the neuron(s) of interest (making these rules function similarly to classical Hebbian updates [36], although there are important differences to them, as discussed in [35]). Since a neuron is able to immediately generate a hypothesis given its own internal state, without requiring the active generation of other predictors, and its error can be readily computed after prediction by comparing to the current state of the target neuron state, the weight updates of any predictor layer may be computed in parallel to others. This would allow us to allocate dedicated computing cores to particular predictors, or "pieces", of the S-NCN. Observe that the S-NCN does not require activation derivatives in any of its computations (this is neurobiologically more realistic and favorable for specialized hardware implementations).

Furthermore, during the learning process, each context vector $\mathbf{g}_t^\ell$ and its corresponding memory matrix is adjusted according to the following simple contrastive rule:

$$\mathbf{g}_{t+1}^\ell = \mathbf{g}_t^\ell + \eta_e\mathbf{d}^\ell - \eta_g(\mathbf{g}_t^\ell - \frac{1}{t-1}\sum_{j=1}^{t-1}(\mathbf{g}_j^\ell)), \text{ and, } \Delta\mathbf{M}^\ell = \mathbf{g}_{t+1}^\ell * (\mathbf{t}_j)^T \qquad (4)$$

where $\eta_e$ modulates a long-term memory update using the current perturbation to be applied to layer $\ell$. $\eta_g$ controls the repulsion term, which "pushes" context codes away from each other (for diversity). These adapted codes, which influence inter-neuronal competition in a task-sensitive manner, could be viewed as a simplification of distributed temporal context [37], where contiguity, i.e., recall/generation of one item is influenced by the presence of another, is introduced into S-NCN distributed processing.

The pseudocode illustrating how the elements described so far are combined in an S-NCN system is presented in Algorithms 1 and 2. The transmission of bottom-up and top-down errors in the S-NCN is motivated by the theory of predictive processing [38, 39, 40, 41, 42, 43, 44] and classical work on interactive networks [34, 45, 46], where models undergo a settling process to process input stimuli more than once (see Appendix for intuition). Though this requires extra computation, the process endows the network with desirable properties, e.g., the ability to conduct constraint satisfaction [47, 48]. By using a multi-step, laterally-competitive processing scheme, the S-NCN is able to "select" subnetworks (portions of neurons) for specific tasks, reducing representational overlap and,

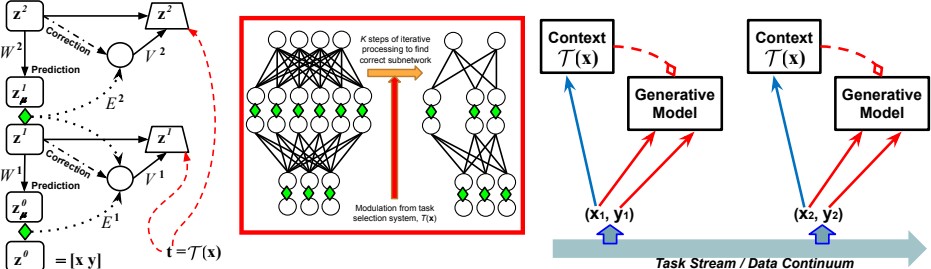

Figure 1: (Left) The sequential neural coding network is shown processing an input $(\mathbf{y}, \mathbf{x})$ over one step, given a task context $\mathbf{t}$ (produced by the task selection model $\mathcal{T}(\mathbf{x})$), overriding the current $\mathbf{z}^\ell$ after error-correction and application of lateral inhibition matrices $\{\mathbf{V}^1, \mathbf{V}^2\}$. Inside the red box is shown one possible emergent subnetwork. Green diamonds indicate error units. (Right) The full complementary neural system shown processing patterns from a data continuum.

ultimately, forgetting. This selection is driven by the task pointer $\mathbf{t}$, produced by the task selector, $\mathcal{T}(\mathbf{x})$, the final piece of the S-NCN system, which we describe in Section 3.3.

### 3.2.3 Objective Function

During training, when presented with stimulus $(\mathbf{y}_i, \mathbf{x}_i, \mathbf{t}_i)$, the S-NCN adjusts its internal states and synapses so that the output $\mathbf{z}_\mu^0$ of layer 1 ($P_1$) is as close as possible to $(\mathbf{y}_i, \mathbf{x}_i)$. It does this by minimizing *total discrepancy* (ToD) [49] – a measure of its total internal disorder,which is the sum of all mismatches between layerwise guesses and actual states. Formally, the ToD for an S-NCN is:

$$\mathcal{L}(\Theta) = \sum_{\ell=0}^{L-1} \mathcal{L}^\ell(\Theta^\ell), \text{ where } \mathcal{L}^\ell(\Theta^\ell) = \frac{1}{2}(||\mathbf{z}^\ell - \mathbf{z}_\mu^\ell||_2)^2. \tag{5}$$

$\Theta$ contains all of the synaptic parameters, i.e., $\Theta = \{\mathbf{W}^1, \mathbf{E}^1, \cdots, \mathbf{W}^L, \mathbf{E}^L\}$ and $\Theta^\ell = \{\mathbf{W}^\ell, \mathbf{E}^\ell\}$.

The above loss decomposes the problem of credit assignment in the S-NCN into sub-problems with each sub-problem focusing on the comparison between the prediction $\mathbf{z}_\mu^\ell$ made by layer $\ell + 1$ and the actual state value $\mathbf{z}^\ell$ of layer $\ell$. The resulting updates to each state $\mathbf{z}^\ell$, as well as relevant synapses, will then depend on a bottom-up transmitted error signal and the top-down influence of the mismatch with the expectation of the predictor immediately above [49, 35]. While we have motivated aspects of our model from a neuro-cognitive perspective, the error units and weight updates can be derived from the total discrepancy function above [35] (and cast as approximately minimizing free energy [50]).

### 3.3 The Neural Task Selection Model

Earlier, we described the S-NCN as taking in a task signal $\mathbf{t}$ that drives the system's task context memory $\mathbf{g}^\ell$ for layer $\ell$ (i.e., the routine $getContexts(\mathbf{t}, \mathcal{M})$ in Algorithm 1). To create this task pointer, we design a second neural circuit that we call the task selector $\mathcal{T}(\mathbf{x})$ – this will allow the S-NCN to automatically decide when a new task has been encountered and to determine, at test time, what task observed data points belong to. The motivation behind $\mathcal{T}(\mathbf{x})$ comes from the neuroscientific idea that the basal ganglia plays the role of information routing (among other roles), which serves as a form of task selection [51, 52]). In other words, it selects/enables various cognitive programs stored in other cortical regions [5]. As a result, we develop a type of complementary learning system (CLS) (different than that of [53], which models an interaction between the hippocampus and neocortex), which pairs our cortex-like model with a basal ganglia-like model. We refer to our task selector $\mathcal{T}(\mathbf{x})$ as the "functional neural basal ganglia" (FNBG) in order to emphasize that the actual basal ganglia in the human brain is more complex and does more than what our computational model does. Our FNBG model, built with competitive learning, has two roles: 1) task shift detection - deciding whether data from an input stream indicates the presence of a new task, and 2) task recognition - deciding whether incoming data requires switching to an existent task context or creating a new one.

**Task Shift Detection:**  In order to detect the occurrence of a new task while processing data from the pattern stream, $\mathcal{T}(\mathbf{x})$ utilizes the output error neurons of the generative model described earlier to detect spikes in their values which might indicate distributional drift. Specifically, $\mathcal{T}(\mathbf{x})$ maintains an

exponentially weighted running mean $\mu_\mathcal{L}$ and variance $\sigma_\mathcal{L}^2$ of the norm of the cortex model's label error neurons $\mathbf{e}_y^0$. The necessary statistics are calculated as follows:

$$\Delta = ||\mathbf{e}_y^0||_2^2 - \mu_\mathcal{L}(t-1) \tag{6}$$

$$\mu_\mathcal{L}(t) = \mu_\mathcal{L}(t-1) + \eta\Delta \tag{7}$$

$$\sigma_\mathcal{L}^2(t) = (1-\eta)\sigma_\mathcal{L}^2(t-1) + \eta(\Delta)^2 \tag{8}$$

with $\eta = 0.1$ (determined by preliminary experiments). Using the above dynamic statistics, a shift is detected by determining if the following inequality evaluates to true (repeatedly for a series of 5 consecutive batches): $\mu_\mathcal{L}(t) > \mu_\mathcal{L}(t-1) + 2\sqrt{\sigma_\mathcal{L}^2(t-1)}$. Upon detection of a boundary, we suppress the check until 1000 samples have been seen after the last detected shift (creating a refractory period to allow the competitive learning model, described next, to acquire enough data to learn).

**Task Recognition through Competitive Learning:** To conduct task recognition, $\mathcal{T}(\mathbf{x})$ first randomly projects the input $\mathbf{x}$ to a low-dimensional space ("key") $\mathbf{k} = \mathbf{R} \cdot \mathbf{x}$ ($\mathbf{R}$ is initialized from a Gaussian distribution). We then update a rolling average estimate of the streaming input using this generated key as follows: $\mathbf{k}_\mu = (1-\tau)\mathbf{k}_\mu + \tau\mathbf{k}$ (with $\tau = 0.65$). Finally, with the matrix $\mathbf{Q}$, the FNBG maps this rolling representation $\mathbf{k}_\mu$ to a decision as to which task context the S-NCN generative cortex is to utilize, i.e., $\hat{\mathbf{t}} = \text{BKWTA}(\mathbf{Q} \cdot (\mathbf{k}_\mu/||\mathbf{k}_\mu||)_2, K = 1)$.The task pointer is then used to retrieve contexts $\{\mathbf{g}_t^1, ..., \mathbf{g}_t^L\}$ where $\mathbf{g}^\ell = \mathbf{M} \cdot \hat{\mathbf{t}}$ (implementing $getContexts(\mathbf{t} = \hat{\mathbf{t}}, \mathcal{M})$).

To update the FNBG weights, while also avoiding catastrophic interference in $\mathcal{T}(\mathbf{x})$ itself, we propose a biologically-inspired learning rule based on competitive learning. Specifically, we develop what we call "guided competitive learning", since during the act of processing a stream of certain samples from the task that we know that we are operating on, we also know which neuron out of a set of $T-1$ task output neurons should be selected. This leads to the following update rule formally defined as:

$$\Delta\mathbf{Q} = -(\rho\mathbf{t}) \cdot (\mathbf{k}_\mu/||\mathbf{k}_\mu||_2)^T \tag{9}$$

where $\rho$ is the competitive weight adjustment factor (a value we found works well in the range of $[0.5, 1]$). Note that $\mathbf{t} = \hat{\mathbf{t}}$ recovers an unsupervised classical competitive Hebbian learning. However, we force the model to a specific task pointer value by using $\mathbf{t}$, the one-hot encoding of the dynamic integer variable $t$ maintained by the FNBG itself, initialized as $t = 0$. Every time a task shift is detected according to Equations 6-8, this dynamic variable is incremented by one, i.e., $t \leftarrow t + 1$.

For both task recognition and the FNBG's synaptic update, note that the rolling representation $\mathbf{k}_\mu$ is normalized by its Euclidean norm so that we may utilize a vectorizable form of competitive learning based on dot products (taking advantage of GPU-based matrix multiplication). In essence, we take the (normalized) rolling average representation of the input stream for a given task, compute its dot-product with all currently-available task weight vectors, and choose the dot product with maximal value as the winner. Finally, we re-normalize the matrix $Q$ by its column Euclidean norms after each of its updates, i.e., $Q[:, i] = (Q[:, i] + \Delta Q[:, i])/||Q[:, i]||_2$ where $Q[:, i]$ indicates the extraction of all values in column $i$ from $Q$ (this normalization is similar to that of adaptive resonance theory [54]).

### 3.4 Putting It All Together: A Complementary System

At a high level, given the above, the full S-NCN complementary system, depicted in Figure 1 (Right), consists of: 1) a task selection model (inspired by the information routing/executive control behavior of the basal ganglia [5]) which creates the task contexts that laterally inhibit/gate the activities of the generative S-NCN, and 2) a generative model that learns to predict inputs/labels given a task context. In essence, the FNBG $\mathcal{T}(\mathbf{x})$ takes in $\mathbf{x}_j$ to produce $\hat{\mathbf{t}}$ (a one-hot representation of a task pointer $t_i$) which is then fed into the generative S-NCN (along with $\mathbf{x}$ and $\mathbf{y}$) to compute predictions. [3]

## 4 Experiments

**Simulation Setup:** In our experiments, we train models with three hidden layers, whether they be multilayer perceptrons (MLPs) or S-NCNs and compare against baselines from the literature. All models were restricted to contain (a maximum of) 500 units per layer. For the S-NCN, weights were initialized from a Gaussian distribution scaled by each layer's fan-in and were optimized using

---

[3]Please see the Appendix where we summarize symbols, notation, and abbreviation used in this work.

stochastic gradient descent with learning rate of $\lambda = 0.01$. Baseline models were trained on each task for 40 epochs while the S-NCN only made a single pass. The output layer for each MLP was a maximum entropy classifier and the objective was to minimize Categorical cross entropy – in the S-NCN, this was encoded in its label error neurons $\mathbf{e}_y^0$. (See Appendix for experimental details, computing infrastructure, hyper-parameter details, and code details.)

**Evaluation Metrics:** To measure model generalization over the sequence of tasks, we make use of the resulting task matrix $R$ (as in [55]), an $N \times N$ matrix of task accuracy scores (normalized to $[0, 1]$), where in this study $N = T$. We measure average accuracy (ACC) (mean performance across tasks) and backward transfer (BWT). BWT measures the influence that learning a task $T_t$ has on the performance of task $T_k < T_t$. A positive BWT indicates that a learning task $T_t$ increases performance on preceding task $T_k$. As such, higher BWT is better and a strongly negative BWT means there is stronger (more catastrophic) forgetting. Mean and standard deviation (10 trials) are reported for ACC and mean (10 trials) for BWT (see Appendix for its standard deviation). The formulas for ACC, BWT, and a new set of metrics we created to analyze memory retention, are provided in the Appendix.

## 4.1 The Multi-Dataset Task Stream

To start, we tested the S-NCN model on a complex task sequence composed of several learning benchmarks [4].We create task sequences by breaking apart MNIST (*M*), Fashion MNIST (*FM*), and Google Draw (*GD*) each into two "sub-tasks" (e.g., for MNIST, *M1* and *M2*), or portions of data with a particular subset of the original dataset's classes. See the Appendix for details on sub-tasks/task sequence creation. In Table 1, we present two task orderings (Ordering #1 is "High Color Sim." and Ordering #2 is "Low Color Sim.") each under two conditions: sub-tasks that have 1) an equal number of classes (5 each), and 2) an unequal number of classes (number of classes was chosen randomly, omitting 5 as an option). The number of classes was sampled once and held constant for all trials.

We evaluate our proposed S-NCN system (as well as four variations of it in the Appendix) – hyperparameters were $\beta = 0.05$, $K = 10$, $\eta_g = 0.9$, $\eta_e = 0.01$, $\alpha = 0.98$). The baselines include an MLP trained only with backprop (Backprop), Elastic Weight Consolidation (EWC) [19], the Mode-IMM method [56], the Md-IMM method combined with either DropTransfer (DT+Md-IMM) or both L2-Transfer and DropTransfer (L2-DT-Mode-IMM) [56], and the competitive model, hard attention to task (HAT) [57]. For each baseline, we tuned hyper-parameters based on their accuracy on each task's development set. See Appendix for additional baseline results.

**Discussion:** Results are reported in Table 1 (see Appendix for more results). Each simulation was run 10 times (each trial used a unique seed).As observed in our results, **we see that the S-NCN outperforms all baselines consistently, in terms of ACC and BWT, exhibiting improved memory retention over modern, backprop-centric baselines** (and, in the Appendix, our expanded results show that the FNBG-driven lateral inhibition is key to improving memory retention the most). This result is robust across both task sequences and equal/unequal class settings. Meta-parameters for the S-NCNs were only tweaked minorly, with the same values used in all scenarios. The observation that lateral inhibition improves the neural computation (in our model) also corroborates the result of [48].

## 4.2 Continual Learning Benchmarks

To connect our model to current learning results, we experimented with a wide swath of approaches on three benchmarks – Split MNIST, Split NotMNIST, and Split Fashion MNIST (FMNIST). Furthermore, we compare to multi-head (below dashed line in Table 2) and single-head models (above dashed line). We compare the S-NCN to both replay/rehearsal and non-replay methods: naïve rehearsal with memory (NR+M), EWC, synaptic intelligence (SI) [58], MAS [59], Lwf [60], GEM [61], ICarl [62], Lucir [63], and Mnemonics [64] (additional baselines can be found in the Appendix).

**Discussion:** In Table 2, we report ACC and BWT, averaged over 10 trials, offering a comprehensive comparison of methods and demonstrating that, for all three benchmarks, **the proposed S-NCN outperforms all of them in terms of ACC**, and nearly all in terms of BWT (and on par with GDumb and GEM, the difference in BWT being negligible) demonstrating the power afforded by designing

---

[4]S-NCN is single head model as opposed to HAT ,GEM and others that use multiple-head

Table 1: Generalization metrics (10 trials) for sequence orderings # 1 & #2 (higher values are better).

| | Ordering #1: $\{M1, M2, GD1, FM1, FM2, GD2\}$ (High Color Sim.) | | | |
| | Equal | | Unequal | |
| | ACC | BWT | ACC | BWT |
|---|---|---|---|---|
| Backprop | $0.241 \pm 0.050$ | $-0.759 \pm 0.030$ | $0.185 \pm 0.048$ | $-0.791 \pm 0.048$ |
| EWC | $0.280 \pm 0.023$ | $-0.714 \pm 0.030$ | $0.185 \pm 0.046$ | $-0.726 \pm 0.039$ |
| Md-IMM | $0.521 \pm 0.027$ | $-0.392 \pm 0.023$ | $0.480 \pm 0.039$ | $-0.240 \pm 0.040$ |
| DT+Md-IMM | $0.530 \pm 0.024$ | $-0.387 \pm 0.021$ | $0.551 \pm 0.042$ | $-0.220 \pm 0.042$ |
| L2+DT+Md-IMM | $0.532 \pm 0.025$ | $-0.237 \pm 0.027$ | $0.520 \pm 0.040$ | $-0.240 \pm 0.045$ |
| HAT | $0.550 \pm 0.019$ | $-0.211 \pm 0.020$ | $0.492 \pm 0.031$ | $-0.231 \pm 0.036$ |
| S-NCN (ours) | $\mathbf{0.716 \pm 0.013}$ | $\mathbf{-0.031 \pm 0.017}$ | $\mathbf{0.713 \pm 0.011}$ | $\mathbf{-0.041 \pm 0.012}$ |
| | Ordering #2: $\{GD2, M1, FM2, M2, GD1, FM1\}$ (Low Color Sim.) | | | |
| Backprop | $0.303 \pm 0.030$ | $-0.644 \pm 0.037$ | $0.287 \pm 0.043$ | $-0.671 \pm 0.043$ |
| EWC | $0.303 \pm 0.031$ | $-0.643 \pm 0.033$ | $0.291 \pm 0.039$ | $-0.663 \pm 0.047$ |
| Md-IMM | $0.584 \pm 0.027$ | $-0.091 \pm 0.030$ | $0.533 \pm 0.034$ | $-0.230 \pm 0.036$ |
| DT+Md-IMM | $0.591 \pm 0.020$ | $-0.088 \pm 0.032$ | $0.528 \pm 0.036$ | $-0.211 \pm 0.039$ |
| L2+DT+Md-IMM | $0.630 \pm 0.029$ | $-0.076 \pm 0.030$ | $0.551 \pm 0.037$ | $-0.201 \pm 0.041$ |
| HAT | $0.596 \pm 0.026$ | $-0.114 \pm 0.029$ | $0.563 \pm 0.031$ | $-0.210 \pm 0.044$ |
| S-NCN (ours) | $\mathbf{0.721 \pm 0.014}$ | $\mathbf{-0.042 \pm 0.013}$ | $\mathbf{0.667 \pm 0.011}$ | $\mathbf{-0.097 \pm 0.013}$ |

Table 2: Generalization metrics (10 trials) for Split MNIST, Split Fashion MNIST (FMNIST) and Not-MNIST benchmarks. Above dashed line are multi-head models & below are single-head models.

| | MNIST | | Fashion MNIST | | NotMNIST | |
| | ACC | BWT | ACC | BWT | ACC | BWT |
|---|---|---|---|---|---|---|
| EWC | $0.760 \pm 0.030$ | $-0.210$ | $0.739 \pm 0.020$ | $-0.201$ | $0.790 \pm 0.020$ | $-0.176$ |
| VCL | $0.980 \pm 0.210$ | $-0.003$ | $0.980 \pm 0.20$ | $-0.002$ | $0.953 \pm 0.003$ | $-0.004$ |
| IMM | $0.951 \pm 0.018$ | $-0.007$ | $0.950 \pm 0.013$ | $-0.005$ | $0.925 \pm 0.011$ | $-0.006$ |
| HAT | $0.972 \pm 0.011$ | $-0.040$ | $0.968 \pm 0.011$ | $-0.004$ | $0.942 \pm 0.009$ | $-0.005$ |
| GEM | $0.922 \pm 0.110$ | $\mathbf{+0.001}$ | $0.930 \pm 0.12$ | $+0.001$ | $0.919 \pm 0.021$ | $-0.003$ |
| A-GEM | $0.950 \pm 0.09$ | $\mathbf{+0.001}$ | $0.955 \pm 0.11$ | $+0.001$ | $0.925 \pm 0.020$ | $-0.002$ |
| ER | $0.938 \pm 0.06$ | $\mathbf{-0.002}$ | $0.945 \pm 0.10$ | $-0.004$ | $0.927 \pm 0.016$ | $-0.004$ |
| EWC | $0.190 \pm 0.030$ | $-0.357$ | $0.199 \pm 0.06$ | $-0.350$ | $0.186 \pm 0.020$ | $-0.361$ |
| NR+M | $0.950 \pm 0.470$ | $-0.100$ | $0.948 \pm 0.380$ | $-0.090$ | $0.880 \pm 0.028$ | $-0.103$ |
| SI | $0.197 \pm 0.110$ | $-0.367$ | $0.198 \pm 0.100$ | $-0.370$ | $0.161 \pm 0.030$ | $-0.370$ |
| MAS | $0.195 \pm 0.290$ | $-0.340$ | $0.180 \pm 0.250$ | $-0.340$ | $0.178 \pm 0.060$ | $-0.341$ |
| Lwf | $0.846 \pm 0.340$ | $-0.120$ | $0.875 \pm 0.300$ | $-0.130$ | $0.626 \pm 0.091$ | $-0.130$ |
| ICarl | $0.940 \pm 0.410$ | $-0.100$ | $0.960 \pm 0.400$ | $-0.110$ | $0.887 \pm 0.102$ | $-0.109$ |
| Lucir | $0.940 \pm 0.310$ | $-0.103$ | $0.950 \pm 0.340$ | $-0.110$ | $0.935 \pm 0.093$ | $-0.101$ |
| GDumb | $0.978 \pm 0.09$ | $\mathbf{-0.005}$ | $0.973 \pm 0.09$ | $-0.006$ | $0.940 \pm 0.080$ | $-0.004$ |
| Mnem | $0.960 \pm 0.320$ | $-0.091$ | $0.968 \pm 0.300$ | $\mathbf{+0.007}$ | $0.950 \pm 0.071$ | $-0.011$ |
| S-NCN | $\mathbf{0.981 \pm 0.300}$ | $-0.005$ | $\mathbf{0.982 \pm 0.400}$ | $-0.003$ | $\mathbf{0.957 \pm 0.400}$ | $\mathbf{-0.004}$ |

models with stronger grounding in neurobiology. Furthermore, it is promising to see that the S-NCN outperforms/matches performance with not only the single-head models but also with multi-head models (except GEM), which enjoy an easier version of the problem, i.e., they can utilize a different classifier per task. Finally, note that the S-NCN, due to the FNBG-driven competition, learns to compose task contexts in a data-dependent manner. Desirably, the S-NCN is a single-head model, meaning that it does not grow out a separate softmax classifier per task (as in multi-headed models, e.g., HAT, IMM, GEM), which means that it tackles the harder form of the forgetting problem, learning representations that preserve knowledge across disjoint tasks. Furthermore, our model is online, whereas models such as IMM or SI require multiple passes per dataset, and does not require validation data in order to run an expensive neural architecture search (NAS) as in [65].

## 4.3 Discussion: On Model Limitations

Although our results with respect to catastrophic forgetting are promising, the S-NCN does have several limitations that are important to discuss. First and foremost, the S-NCN attempts to learn a generative model of the joint distribution over inputs and labels, i.e., $p(\mathbf{y}, \mathbf{x})$, which has (classically) been shown to be a much more difficult problem compared to directly learning a conditional mapping between inputs and labels [66]. Second, to learn $p(\mathbf{y}, \mathbf{x})$, the S-NCN adapts to data through an expectation-maximization process where, given an input, it must iteratively infer a maximum a

posteriori (MAP) estimate of its latent neural activities over a $K$-step stimulus window. Despite the fortunate fact that, for the benchmarks studied in this work, values for $K$ were fairly low (only 10 to 20 steps were needed per sample/mini-batch), for more complex data types, such as natural images with multiple objects and background scenery, the value of $K$ will quite likely need to be much higher, increasing the S-NCN's computation time when conducting its online inference. We remark that this drawback could be mitigated by integrating mechanisms to support amortized inference, e.g., as in predictive sparse decomposition [67], and by designing custom software/hardware implementations that exploit the S-NCN's layer-wise parallelism (which would also work in asynchronous settings) in both its inference and weight updating cycles. Future work should explore adapting the S-NCN to only learning a conditional mapping as opposed to a full joint distribution as well as develop mechanisms for pre-training the generative side of the system (which would allow freezing of the generative synapses and only require updating discriminative ones – this could reduce the value of $K$ for more complex sensory inputs). While the S-NCN's objective of minimizing ToD is important for breaking free of backprop and its limitations, offering benefits such as resolving weight transport, eschewing the need for derivatives, facilitating local learning without the need for a global feedback pathway, inferring richer stateful representations, it also creates a more challenging optimization problem, i.e., the model must match the values imposed by input data as well as ensure that its internal activities and local predictions are aligned. While the full complementary system largely mitigates catastrophic interference, this primarily benefits measurements of BWT and could potentially damage the model's per-task performance on more complex datasets, i.e., the main diagonal of task matrix $R$. Since we do not impose any distributional assumptions over the latent activities (such as a clean Gaussian prior as in variational autoencoders), if the S-NCN's estimated value of the latent activities is far from their true posterior distribution, then the S-NCN might experience sub-optimal performance in complex setups. Even though all continual learning systems suffer from this issue (especially most modern-day ANN-based ones), our model's dual optimization nature could experience this problem more frequently. We believe that integrating memory-aware retrieval driven by (a brain-like form of) replay, which would exploit the S-NCN's generative nature, could be a plausible solution moving forward (helping the system come closer to its true posterior and avoiding bad local optima).

### 4.4   On Negative Societal Impacts

The potential negative societal impact of the proposed S-NCN complementary system is indirect – while the model and algorithmic framework we develop is foundational in nature, it could potentially affect the myriad of applications/systems currently in development today or those to be developed in the future. As a result, at best, the same negative consequences that result from using backprop-based ANNs are still present when using the S-NCN training process instead. At worst, given that we have presented promising results across several lifelong learning benchmarks, the S-NCN could facilitate the development of better-performing robotic agents that might be used in military applications that might result in the loss of life, such as drones. Despite the benefits that S-NCN offers to the statistical learning and cognitive neuroscience communities, one should consider the drawbacks of building powerful neural systems to drive applications. To safeguard against its potential negative impact, we emphasize that developing an ethical framework to guide the design and training of general intelligent systems will be important for ensuring safe integration into human society.

## 5   Conclusion

In this paper, we proposed the sequential neural coding network (S-NCN), an interactive generative model, and its local learning procedure for lifelong machine learning. As demonstrated on several benchmarks and setups, this model retains the knowledge it acquires from prior tasks when learning new ones in the face of task data streams, primarily when lateral inhibition, driven by a self-organizing task selection model, sharpens neural activities within its layers. As a result, this work marks an important step towards developing brain-inspired agents that are capable of robustly combating catastrophic forgetting, especially ones that do not require task descriptors to guide the learning process, offering a promising pathway towards achieving the ultimate goal of designing agents that act and adapt in ways more similar to animals and humans.

## Acknowledgements

We would like to thank Alexander Ororbia (Sr.) for useful discussions related to cognitive types, a concept that served as key motivation for the task contexts that drove lateral inhibition in this work.

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
