# Lifelong Neural Predictive Coding: Learning Cumulatively Online without Forgetting (Supplementary Material)

**Alexander G. Ororbia**
Rochester Institute of Technology
Rochester, NY 14623, USA
ago@cs.rit.edu

**Ankur Mali**
University of South Florida
Tampa, FL 33620, USA
ankurarjunmali@usf.edu

**C. Lee Giles**
The Pennsylvania State University
State College, PA 16801, USA
clg20@psu.edu

**Daniel Kifer**
The Pennsylvania State University
State College, PA 16801, USA
duk17@psu.edu

## Computational Resource Setup:

All experiments were performed with 128GB RAM on an intel Xeon server with 3.5GHZ processors, consisting of 4 1080Ti GPUs. Any of our models can easily fit into single 1080Ti GPUs – a multi GPU setup was only used to speed up the computation. All models are coded in Tensorflow 2.2 and we only use basic parallelism provided by the Tensorflow library to speed up computation.

**Experimental Code:** Code to support this work can be found at the following URL/repository: https://github.com/ago109/lifelong_pc.git.

## Theoretical Foundations and Technical Details of Sequential Neural Coding

In this appendix section, we present definitions and mathematical derivations, answers to key questions, and further detail core intuitions related to sequential neural coding (i.e., the S-NCN system presented in the main paper).

**Definition Table:** In Table 1, we explain what each mathematical symbol/operation/abbreviation in the main paper represents.

**Derivation of State & Weight Updates:** As mentioned in the main paper, the S-NCN's generative circuit minimizes an objective function known as total discrepancy (ToD) when it is presented with input stimuli $(\mathbf{x}_i, \mathbf{y}_i)$. The ToD is formally:

$$\mathcal{L}(\Theta) = \sum_{\ell=0}^{L-1} \frac{1}{2} (||\mathbf{z}^\ell - \mathbf{z}_\mu^\ell||_2)^2 = \sum_{\ell=0}^{L-1} \frac{1}{2} \sum_j \left( \mathbf{z}^\ell[j] - \mathbf{z}_\mu^\ell[j] \right)^2 \tag{1}$$

where $\mathbf{z}^\ell[j]$ means that we extract the $j$-th element of vector $\mathbf{z}^\ell$ (and we have simplified the expression by squaring the square root operator of the L2 norm, giving us a sum of squared dimensions). Since all of the latent states of the generative circuit are continuous, the updates will follow the form of the exact gradient, i.e., differentiation (which would permit the use of gradient descent), to optimize the latent variables and the synaptic weight parameters. Given this, the partial derivative of Equation 1

36th Conference on Neural Information Processing Systems (NeurIPS 2022).

Table 1: Table of key symbol/operator/abbreviation definitions.

| Item | Explanation |
|---|---|
| S-NCN | Sequential neural coding network (model) |
| FNBG | Functional neural basal ganglia (model) |
| $\mathbf{v} \in \mathcal{R}^{D \times 1}$ | A column vector $\mathbf{v}$ of shape $D \times 1$ |
| $\mathbf{M} \in \mathcal{R}^{B \times D}$ | A matrix $\mathbf{M}$ of shape $B \times D$ |
| $\cdot$ | Matrix/vector multiplication |
| $\odot$ | Hadamard product (element-wise multiplication) |
| $(\mathbf{v})^T$ | Transpose of $\mathbf{v}$ |
| $\|\|\mathbf{v}\|\|_2$ | Euclidean norm of $\mathbf{v}$ |
| $\mathbf{x}_j$ | The $j$th data point (image) sampled from task $\mathcal{T}_i$ |
| $\mathbf{y}_j$ | The $j$th data label (one-hot encoded) sampled from task $\mathcal{T}_i$ |
| $P_\ell$ | The $\ell$-th predictor/layer of the S-NCN generative circuit. |
| $J_x$ | Dimensionality of input $\mathbf{x}_j$ |
| $J_\ell$ | Number of neurons in layer $P_\ell$ of the S-NCN generative circuit. |

with respect to any layer of neural activities (or latent state) $\mathbf{z}^\ell$ would be:

$$\frac{\partial \mathcal{L}(\Theta)}{\partial \mathbf{z}^\ell} = \left( \frac{\partial \mathbf{z}_\mu^{\ell-1}}{\partial \mathbf{z}^\ell} \cdot \left( (\mathbf{z}^{\ell-1} - \mathbf{z}_\mu^{\ell-1}) \right) \right) - (\mathbf{z}^\ell - \mathbf{z}_\mu^\ell) \tag{2}$$

$$= \left[ (\mathbf{W}^\ell)^T \cdot (\mathbf{z}^{\ell-1} - \mathbf{z}_\mu^{\ell-1}) \right] \odot \frac{\partial \phi^\ell(\mathbf{z}^\ell)}{\partial \mathbf{z}^\ell} - (\mathbf{z}^\ell - \mathbf{z}_\mu^\ell) \tag{3}$$

$$= (\mathbf{W}^\ell)^T \cdot (\mathbf{e}^{\ell-1}) \odot \frac{\partial \phi^\ell(\mathbf{z}^\ell)}{\partial \mathbf{z}^\ell} - \mathbf{e}^\ell \tag{4}$$

where we notice that the error neurons are derived directly from the ToD objective as well, i.e., $\mathbf{e}^\ell = \frac{\partial \partial \mathcal{L}(\Theta)}{\partial \mathbf{z}_\mu^\ell} = \mathbf{z}^\ell - \mathbf{z}_\mu^\ell$ (allowing us to write Equation 3 in terms of error neurons as in Equation 4). Alternatively, by replacing the term $\frac{\partial \mathbf{z}^{\ell-1}}{\partial \mathbf{z}^\ell}$ with a learnable error matrix $\mathbf{E}^\ell$ instead, Equation 3 can be simplified to the following:

$$\frac{\partial \mathcal{L}(\Theta)}{\partial \mathbf{z}^\ell} \approx \mathbf{d}^\ell = \mathbf{E}^\ell \cdot \mathbf{e}^{\ell-1} - \mathbf{e}^\ell \tag{5}$$

which is a stable derivative-free perturbation $\mathbf{d}^\ell$ (so long as the activation function $\phi^\ell()$ is monotonically increasing) to the neural activities (and as noted in [1], the dampening effect that the activation derivative $\frac{\partial \phi^\ell(\mathbf{z}^\ell)}{\partial \mathbf{z}^\ell}$ would have can be approximated with biologically-plausible dampening functions if needed). The final update to the latent neural activities is then performed using a gradient-ascent like operation, i.e., $\mathbf{z}^\ell(k) = f^\ell(\mathbf{z}^\ell(k-1) + \beta \mathbf{d}^\ell)$ (this was presented in the main paper).

Deriving the updates to the synaptic generative parameters is also done in a similar fashion as above, i.e., by taking the gradient of ToD with respect to $\mathbf{W}^\ell$.

$$\frac{\partial \mathcal{L}(\Theta)}{\partial \mathbf{W}^\ell} \propto \Delta \mathbf{W}^\ell = \frac{\partial \mathcal{L}(\Theta)}{\partial \mathbf{z}_\mu^\ell} \cdot \left( \phi^{\ell+1}(\mathbf{z}^{\ell+1}) \right)^T, \text{ where, } \mathbf{z}_\mu^\ell = \mathbf{W}^\ell \cdot \phi^{\ell+1}(\mathbf{z}^{\ell+1}) \tag{6}$$

$$= (\mathbf{z}^\ell - \mathbf{z}_\mu^\ell) \cdot (\phi^{\ell+1}(\mathbf{z}^{\ell+1}))^T = \mathbf{e}^\ell \cdot (\phi^{\ell+1}(\mathbf{z}^{\ell+1}))^T. \tag{7}$$

If we are using $\mathbf{E}^\ell$ feedback/error matrices (as we do in this paper), we can leverage a simple Hebbian update $\Delta \mathbf{E}^\ell = \alpha \left( \phi^{\ell+1}(\mathbf{z}^{\ell+1}) \cdot (\mathbf{e}^\ell)^T \right)$ [1] (which, if applied to $\mathbf{E}^\ell$ every time that Equation 7 is applied to $\mathbf{W}^\ell$, allows $\mathbf{E}^\ell$ to converge to the approximate transpose of $\mathbf{W}^\ell$). Much as was done for the states, synaptic weight matrices are updated via gradient ascent: $\mathbf{W}^\ell = \mathbf{W}^\ell + \lambda \Delta \mathbf{W}^\ell$ and $\mathbf{E}^\ell = \mathbf{E}^\ell + \lambda \Delta \mathbf{E}^\ell$ ($\lambda$ is the learning rate/step size).

**How would a model with symmetric connections behave?** A model without separate feedback connections (in contrast to the S-NCN we experiment that uses asymmetric forward/feedback weights) would behave quite similarly yet favorably offer a reduction in memory cost (one does not need to store separate feedback/error matrices in memory). In other words, one could certainly swap out $\mathbf{E}^\ell$ with $(\mathbf{W}^\ell)^T$ if this memory cost reduction was desired/necessary. However, by utilizing separate learnable

feedback synapses, the S-NCN in the form presented in this study resolves the weight transport problem, a well-known biological criticism of backprop where error/teaching information is carried backwards along the same synapses that were used to forward propagate information.Interestingly enough, in preliminary experimentation, we found that using separate learnable feedback synapses improved the generative modeling/reconstruction ability of the generative cortex (particularly in the online case). Although we will investigate this effect in future work, we note this change in generative performance did not really seem to impact the classification accuracy (arguably because discrimination is easier than generation).

**Initializing Latent States:** In the S-NCN's generative circuit, there are several layers of neural activities that are not clamped to data, e.g., $\mathbf{z}^1, \mathbf{z}^2, ..., \mathbf{z}^L$. These activity vectors, as mentioned in the main paper, are initialized to zero vectors (i.e., $\mathbf{z}^\ell = \mathbf{0}$) before they are updated/modified by the message passing that occurs over $K$ steps of processing input stimuli $(\mathbf{x}, \mathbf{y})$. While these initially zero vectors will eventually become non-zero vectors, particularly after a minimum of $K = L$ steps (for example: after $k = 1$, $\mathbf{z}^1$ will be non-zero given that the error neurons in layer $0$ will be non-zero and thus the layer $1$ state perturbation vector $\mathbf{d}^1$ will contain non-zero entries; after $k = 2$, $\mathbf{z}^2$ will contain non-zero values, and so on and so forth), it is entirely possible to randomly initialize these states with non-zero vectors (though it is recommended to keep the magnitude of the randomly chosen initial numbers relatively small). We leave investigation of alternative initialization schemes for future work.

**Relationship to Surrogate Gradients:** A particular line work that shares interesting relationships with the generative circuit of the S-NCN is that of surrogate gradients, such as decoupled neural interfaces (DNIs) [2, 3]. In essence, this class of methods aims to resolve one of backprop's central issues – the update-locking problem (where updates to one layer's synaptic parameters must for other layer's updates to be computed as error/teaching information is backwards propagated down a serial feedback pathway). The key module driving this class of methods is the introduction of a gradient predictor, which can be adapted/taught to approximate actual gradients as produced by backprop, ultimately, after training the predictor's well enough removing the need for backprop later in training and permitting parallel, asynchronous updates to be made to deep, even recurrent, network architectures. In contrast to surrogate gradient-based approaches, the S-NCN works to compute synaptic updates without resorting to predicting gradients given that its generative circuit is naturally layer-wise parallelizable. In effect, each layer-wise prediction is made independently of the others (unlike the typical forward passes in modern-day deep networks) and the synaptic updates for each layer (both forward and error/feedback synapses) can be made without others having been computed/completed. This opens the door to potential parallel asynchronous implementations of the S-NCN that could drastically speed up its computation further. In contrast to DNIs, the S-NCN's generative cortex does not require training gradient predictors (DNIs typically require access to true gradients provided by backprop in order to train them properly) and, without incurring synthetic approximation errors (as in a fully-unlocked network using DNI, where now even the layer activities require additional modules to be trained to predict actual layer-wise activities) furthermore, resolves both update and forward locking problems. Crucially, the S-NCN's updates are biologically plausible – it only requires simple (multi-term) Hebbian updates for the generative cortex and competitive Hebbian updates for the basal ganglia.

**General S-NCN System Process Intuition:** From a high-level intuitive point-of-view, the S-NCN system described in the main paper is composed of two complementary neural circuits: 1) an interactive generative (cortical) circuit that learns to predict its input stimuli (pixel images and their respective labels), and 2) the functional neural basal ganglia (FNBG) which is a specialized circuit that learns to group pixel images into unique "task" contexts. When presented with a sample or mini-batch at any point within the task stream being sampled, the S-NCN does the following:

1. The FNBG task selection model determines if the currently sampled data is coming from the same task that the S-NCN has currently been processing or if it comes from either new/different task. (Note, not mentioned in the paper, the FNBG can also determine that the current data belongs to a task that it has previously seen by letting its set of neurons compete and determining if the winner has a very high dot product score – usually checked against a threshold). Before letting the S-NCN generative circuit process/adapt to current data, if the FNBG determines that the data is coming from a novel task, it will create a new task context memory $\mathbf{g}_t^\ell$ for each $\mathbf{M}^\ell$ to subsequently drive the S-NCN as it processes current data.

2. Given the task context provided by the FNBG, the S-NCN generative circuit will then process the current data over a fixed stimulus window (or for $K$ discrete time steps), adjusting its synapses so that it may better generate the input sensory sample better in the future as well as predict its label. Note that the update to the S-NCN's synapses will also trigger an update to the task context memory vectors as well as an update to the FNBG's synapses (which are themselves adjusted through competitive Hebbian learning).

**Task Shift Detection Intuition:** To provide further intuition as to how the FNBG task selector operates when detecting the occurrence of a novel task, i.e., "task shift detection", we explain what the key equations presented in the main paper are doing:

1. The FNBG lets the S-NCN generative circuit continue as it normally would and have it make a prediction of both $\mathbf{x}_j$ and $\mathbf{y}_j$. Then, it extracts the label error neuron vector $\mathbf{e}_y^0$ from the generative model and computes its squared Euclidean norm $||\mathbf{e}_y^0||_2^2$.
2. The FNBG maintains two particular scalar parameters $\mu_{\mathcal{L}}$ and $\sigma_{\mathcal{L}}^2$, which are the mean and variance of the squared Euclidean norm of the S-NCN's label error neuron activities. The key Equations 7-9 in the main paper depict how the FNBG updates these mean $\mu_{\mathcal{L}}(t)$ and variance $\sigma_{\mathcal{L}}^2(t)$ parameters (based on an exponentially weighted moving mean and variance).
3. Once the FNBG updates these two statistical parameters, it then performs a check of its current new mean $\mu_{\mathcal{L}}(t)$ against the previous value $(t-1)$ of its mean and variance parameters, i.e., $\mu_{\mathcal{L}}(t-1)$ and $\sigma_{\mathcal{L}}^2(t-1)$. This check is specifically (as in the main paper) the following: $\mu_{\mathcal{L}}(t) > \mu_{\mathcal{L}}(t-1) + 2\sqrt{\sigma_{\mathcal{L}}^2(t-1)}$ which says that, if the current mean of the label error neuron activity (at time $t$) is greater than the sum of the previous mean (at time $t-1$) plus two times the previous standard deviation, the S-NCN system has encountered a *shift in task*, which means that, if this inequality evaluates to true, the currently encountered data sample comes from either a novel task or a previously encountered task (but not from the current task). If this task shift inequality evaluates to true, the FNBG will not allow the generative circuit to update its synapses and instead force it to recompute its prediction of the current data using a newly generated task context. Otherwise, if the inequality evaluates to false, then the FNBG will let the generative circuit continue and update its synapses.

**General Design Motivation/Benefits:** The key motivations behind the S-NCN's design are: 1) to develop a neurobiologically-inspired online approach to learn a generative model of $p(\mathbf{x}, \mathbf{y})$ (one motivated by perceptual cortical circuits), 2) to develop an online information routing model (one motivated/inspired by the basal ganglia) that suppresses/drives the neural activities in the generative model of item (1) depending on the task that it decides that the system is operating on – this crucially removes the need for user-provided task descriptors. In addition, by design, the S-NCN: 1) exhibits no need for activation function derivatives, 2) exhibits no update-locking (i.e., it is layer-wise parallelizable), 3) does not require/need a global feedback pathway to drive/facilitate credit assignment (i.e., side-stepping exploding/vanishing gradient problems), and 4) it resolves the weight transport problem by using asymmetric generative and error-correction synapses (those these could be tied/shared to reduce the S-NCN's memory footprint if need be). In contrast to many current lifelong learning approaches, the S-NCN differs in that it is a complementary system that focuses on the relationship between the basal ganglia (as an information router) and (generative/predictive) cortical regions. This is what allows the S-NCN system to offer the advantage of internal, automatic task selection and task boundary detection, which few modern methods provide.

Although the S-NCN shares predictive processing's (PP's) iterative inference/learning (note that [4] focused on PP's auto-association abilities and further note that [1] focused on PP's generative/sampling abilities); its ability to combat forgetting comes from the interaction between the task selector and the generative circuit (the former drives lateral suppression/excitation in the latter). Note that the S-NCN's generative model could be useful for inducing a form of (internal) layer-wise generative replay (each layer could refresh itself in a sleep phase). This we observe is additional, untapped potential for using the S-NCN's learned directed generative model in a spirit similar to the backprop-based model of [5]. We argue that this self-induced form of replay could prove useful for longer task sequences with far more tasks.

Empirically, we note that asymmetric forward and backwards (error) weights were found to work best for NCN systems without activation derivatives in their state updates, yielding fewer inference steps. However, since backwards weights converge to the approximate transpose of the forward ones, using symmetric weights would reduce memory usage (see earlier discussion/section "How would a model

with symmetric connections behave?"). We further remark that, although we do not explore them in this work, fixed, random backward synapses (as in feedback alignment) would work as well, which we found, in preliminary experimentation, reduced/harmed the S-NCN's ACC by only a few points.

**Relationship to Temporal Neural Coding:** Prior related on temporal neural coding [6, 7] has investigated the design and development of predictive coding frameworks for handling the modeling of time-varying data points (such as frames in a video). However, in contrast to the generative circuit in this work, these previous models were either not layer-wise parallelizable [6] or were created under a specialized recurrent formulation of PC for processing data over multiple epochs [7] (such as bouncing ball or digit videos). These models would not be naturally resistant to forgetting like the S-NCN is, specifically given that the S-NCN is a complementary system where one circuit aids the other in learning task-sensitive/dependent sub-networks. Furthermore, this earlier related work [6, 7] focused on unsupervised sequence modeling and did not investigate discriminative forms of learning (e.g., classification) where forgetting occurs, according to our experience, more quickly and its effects are stronger/more easily observed. Note that in [7], although some improved memory retention across the three sequence modeling tasks was observed, forgetting over tasks was still apparent, and any improvement in memory retention could be be viewed more as a pleasant side-effect rather than the result of a mechanism specialized for safe-guarding against forgetting.

## On Weight Update Modulation

As presented in the main paper, the synaptic weight updates for the generative circuit of the S-NCN applies modulation matrices to improve learning stability over time (specifically invoking a form of synaptic scaling – note that these modulation matrices are not meant to mitigate the occurrence of forgetting). The modulation factors are (locally) computed as a function of the synapses as follows:

$$\widehat{\mathbf{m}}_W^\ell = \Sigma_{j=1}^{J_{\ell+1}} \mathbf{W}^\ell[:,j], \text{ and, } \mathbf{m}_W^\ell = \min\left(\frac{\gamma_s \widehat{\mathbf{m}}_W^\ell}{\max(\widehat{\mathbf{m}}_W^\ell)}, 1\right) \tag{8}$$

$$\mathbf{S}_W^\ell = (\mathbf{W}^\ell * 0 + 1) \odot \mathbf{m}_W^\ell \tag{9}$$

where $\mathbf{W}[:,j]$ denotes the extraction of the $j$th column of $W$, $\max(\widehat{\mathbf{m}}_W^\ell)$ returns the maximum scalar value of $\widehat{\mathbf{m}}_W^\ell$, and $\gamma_s = 2$. We note that the first two formulae collapse the forward matrix to a column vector of normalized multiplicative weighting factors and the third formula converts the column vector to a tiled matrix of the same shape as $\mathbf{W}^\ell$. The error weight modulation factor is computed in fashion similar to that of the forward weights:

$$\widehat{\mathbf{m}}_E^\ell = \Sigma_{j=1}^{J_\ell} \mathbf{E}^\ell[:,j], \text{ and, } \mathbf{m}_E^\ell = \min\left(\frac{2\widehat{\mathbf{m}}_E^\ell}{\max(\widehat{\mathbf{m}}_E^\ell)}, 1\right) \tag{10}$$

$$\mathbf{S}_E^\ell = (\mathbf{E}^\ell * 0 + 1) \odot \mathbf{m}_E^\ell \tag{11}$$

where we observe that modulation factors are computed across the pre-synaptic dimension/side of either matrix $\mathbf{W}^\ell$ or $\mathbf{E}^\ell$. The multiplicative modulation terms come from the insight in neuroscience that synaptic scaling, driven by competition across synapses, serves as a global (negative) feedback mechanism for regulating the magnitude of synaptic adjustments [8, 9, 10]. From a practical standpoint, we found that using the above modulation/scaling factors meant we did not have to craft a synaptic normalization scheme (such as in sparse coding schemes, where the columns/rows of a synaptic matrix must be normalized such that they of unit length each time the matrix is updated).

We remark that the modulation factors we introduce could instead be adapted such that they are useful for mitigating forgetting instead, as has been done in other continual learning approaches [11, 12, 13, 14]. This could help to reduce the cost for growing out new task contexts each time a new task is encountered. For example, one could adapt the modulation factor matrices to instead be conditioned on the output of the S-NCN's functional neural basal ganglia (or another type of task selector circuit, such as one that mimics the cognitive control capabilities of the prefrontal cortex).

## On Partial Pattern Completion

In the event that incomplete input $\mathbf{x}_i$ is provided to the S-NCN, i.e., portions of $\mathbf{x}_i$ are masked out by the variable $\mathbf{m} \in \{0, 1\}^{J_0 \times 1}$, as mentioned in the main paper, we may infer the remaining portions

of $\mathbf{x}_i$ by using the relevant output error neurons of the S-NCN and treating the bottom sensory/input layer $\mathbf{z}_x^0$ as a partial latent state. Specifically, we update the missing portions, i.e., $1 - \mathbf{m}$, of $\mathbf{z}^0$ as:

$$\mathbf{z}_x^0 = \mathbf{x} \odot \mathbf{m} + \left( \mathbf{z}_x^0 - \beta \mathbf{e}_x^0 \right) \odot (1 - \mathbf{m}) \tag{12}$$

where $\mathbf{e}_x^0 = \mathbf{z}_x^0 - \mathbf{z}_{\mu,x}^0$ (error neuron signals related to $\mathbf{x}_i$).

## Parameter Optimization Setup and Baseline Details

**S-NCN Optimization:** For the S-NCN, we used SGD with a learning rate of $\lambda = 0.01$ (this rate was only minorly tuned on the validation set of the first task in preliminary investigation) and mini-batches each containing 10 samples. Based on preliminary experiments, the S-NCN, in general, was found to be robust w.r.t. such hyper-parameters. However, the final meta-parameter values used, i.e., $\beta = 0.05$, $K = 10$, $\eta_g = 0.9$, $\eta_e = 0.01$, $\alpha = 0.98$, were obtained by conducting a grid search (using the validation sets to find best generalization). This meant that we searched over the ranges: $\beta = [0.01, 0.1]$, $K = [5, 30]$, $\eta_g = [0.5, 1.0]$, $\eta_e = [0.005, 0.25]$, $\alpha = [0.5, 1.0]$.

**Baseline Descriptions:** The baselines include an MLP trained exclusively with backprop (Backprop), an MLP trained by backprop but regularized by drop-out (Backprop+DO), Elastic Weight Consolidation (EWC) [15], EWC further regularized by drop-out (EWC-DO), the mean incremental moment matching method (IMM) or Mean-IMM [16], the Mode-IMM method [16], the Mean-IMM method combined with either DropTransfer (DT+Mean-IMM) or L2-transfer (L2-Mean-IMM) or both (L2+DT+Mean-IMM) [16], the Md-IMM method combined with either DropTransfer (DT+Md-IMM) or L2-Transfer (L2+Md-IMM) or both (L2-DT-Mode-IMM) [16], and the state-of-the-art competitive model, hard attention to task (HAT) [17]. Furthermore, as mentioned in the main paper, we examine other methods including those based on replay/rehearsal: naïve rehearsal with memory (NR+M), EWC, synaptic intelligence (SI), MAS [18], Lwf [19], GEM [20], ICarl [21], Lucir [22], and Mnemonics [23]. With respect to very modern baselines, we also include, in the main paper, results for the greedy sampler and dumb learner (GDumb) [24, 25], experience replay (ER) [26, 25], and average gradient episodic memory (A-GEM) [27, 25].

**Baseline Meta-parameter Tuning:** For all baselines we take/use the hyper-parameters from each model/algorithm's source work as a starting point and tuned each, using grid search, the batch size, learning rate, number of hidden units in each layer of the target MLP classifier, and optimizer choice. We tuned across the following ranges: learning rate range was $[3e - 5, 0.1]$, the optimizer choice was tuned across the discrete set ["SGD", "momentum", "Adam", "AdamW"], the hidden layer size range was $[128, 512]$, and number of layers range was $[2 - 5]$. For each baseline, we tuned hyper-parameters based on their accuracy on each task's development set (as mentioned in the next sub-section). For IMM, we used the same settings proposed in the original paper as a starting point [16]. However, we found that HAT [17] was quite sensitive to the choice of its two key hyper-parameters: 1) the stability parameter $s_{max}$, and 2) the "compressibility" parameter $c$. After extensive tuning, we used $s_{max} = 450$ and $c = 0.78$. For other baseline-specific hyperparameters, e.g., A-GEM has a gamma and soft-constraint meta-parameter, we used the best-practice values reported in the literature (as we found that these values worked well in general, even after some preliminary experimentation).

**Limitations (Expanded Discussion):** Our model jointly predicts the target label and learns to generate the sensory input, further driven/modulated by a simple complementary neural system that mitigates neural cross-talk. The dual nature of our model/system helps to uncover distributed representations that facilitate robust learning and adaptation over sequences of tasks/datasets. Even though this design scheme provides flexibility and seems to offer many advantages compared to other backprop-based models, it does come with several limitations. Mainly, finding the true posterior distribution over latent neural activities is harder than just learning a forward mapping between inputs and output targets and, notably, it can be expensive to find a good set of neural activity values as the problem complexity increases (notably the $K$ hyper-parameter, which controls the amount of steps taken per data point/mini-batch by the S-NCN to iteratively infer a potential maximum a posterior estimate of its state variables). Currently, the S-NCN conducts inference and learning through a sort of expectation-maximization process and, fortunately, in the problems we studied, the value of $K$ was fairly low (only 10 to 20 steps at most were needed to find useful state values per sample/mini-batch). However, for more complex data types, such as natural images with multiple objects and even background scenery, the value of $K$ will quite likely need to be much higher, increasing

| Model General Hyperparameters | | | |
|---|---|---|---|
| | **MNIST** | **FMNIST** | **NotMNIST** |
| **Model** | *Configuration* | *Configuration* | *Configuration* |
| EWC | lr = 1e-3, SGD
NH = 1024, NL = 2 | lr = 1e-4, SGD+M
NH = 512, NL = 3 | lr = 1e-3, SGD
NH = 1024, NL = 2 |
| VCL | lr = 2e-4, Adam
NH = 512, NL = 3 | lr = 2e-4, SGD+M
NH = 512, NL = 3 | lr = 2e-4, Adam
NH = 512, NL = 3 |
| IMM | lr = 2e-3, SGD+M
NH = 512, NL = 3 | lr = 1e-5, SGD
NH = 512, NL = 3 | lr = 1e-4, SGD+M
NH = 512, NL = 3 |
| HAT | lr = 1e-4, SGD
NH = 512, NL = 3 | lr = 2e-4, SGD
NH = 512, NL = 3 | lr = 2e-4, SGD+M
NH = 512, NL = 3 |
| A-GEM | lr = 1e-4, SGD+M
NH = 512, NL = 3 | lr = 2e-5, SGD
NH = 512, NL = 3 | lr = 2e-4, SGD+M
NH = 512, NL = 3 |
| ER | lr = 2e-4, Adam
NH = 512, NL = 3 | lr = 1e-4, AdamW
NH = 512, NL = 3 | lr = 1e-4, AdamW
NH = 512, NL = 3 |
| EWC | lr = 1e-3, SGD+M
NH = 1024, NL = 2 | lr = 1e-3, SGD+M
NH = 1024, NL = 2 | lr = 1e-3, SGD
NH = 512, NL = 3 |
| NR+M | lr = 1e-4, SGD+M
NH = 512, NL = 3 | lr = 1e-3, SGD+M
NH = 512, NL = 3 | lr = 1e-4, SGD
NH = 512, NL = 3 |
| SI | lr = 1e-3, SGD
NH = 512, NL = 3 | lr = 1e-3, SGD+M
NH = 512, NL = 3 | lr = 1e-3, SGD
NH = 512, NL = 3 |
| MAS | lr = 1e-4, Adam
NH = 512, NL = 3 | lr = 1e-4, Adam
NH = 512, NL = 3 | lr = 2e-4, Adam
NH = 512, NL = 3 |
| Lwf | lr = 1e-3, SGD
NH = 1024, NL = 2 | lr = 1e-3, SGD
NH = 1024, NL = 2 | lr = 1e-3, SGD
NH = 1024, NL = 2 |
| ICarl | lr = 1e-3, SGD+M
NH = 1024, NL = 2 | lr = 1e-3, SGD+M
NH = 1000, NL = 2 | lr = 1e-4, SGD+M
NH = 1000, NL = 2 |
| Lucir | lr = 1e-4, Adam
NH = 512, NL = 3 | lr = 2e-4, AdamW
NH = 512, NL = 3 | lr = 2e-5, AdamW
NH = 512, NL = 3 |
| GDumb | lr = 2e-4, AdamW
NH = 512, NL = 3 | lr = 2e-5, Adam
NH = 512, NL = 3 | lr = 2e-4, AdamW
NH =512 , NL = 3 |
| Mnem | lr = 1e-4, Adam
NH = 512, NL = 3 | lr = 2e-5, Adam
NH = 500, NL = 3 | lr = 2e-4, Adam
NH = 512, NL = 3 |
| S-NCN | lr = 0.0105, SGD
NH = 500, NL = 3 | lr = 0.01, SGD
NH = 500, NL = 3 | lr = 0.011, SGD
NH = 500, NL = 3 |

Table 2: General hyper-parameter values selected from tuning. Models above double horizontal line are multi-head models and models below it are single-head models. "NH" stands for "number of hidden neurons", "NL" stands for "number of hidden layers", "lr" stands for "learning rate", and optimizer choice is either "SGD" for "stochastic gradient descent" ("SGA" means "stochastic gradient ascent"), "Adam", "AdamW", and 'SGD+M" for "SGD with momentum").

the computation time needed to conduct online inference. This drawback could be mitigated by integrating amortized inference, e.g., predictive sparse decomposition [28], and by designing custom software/hardware implementations to exploit the S-NCN's layer-wise parallelism.

Furthermore, the fact that the S-NCN (in its current form/implementation in this study) must solve a dual optimization problem that entails jointly learning to predict the target label and generate the sensory input (image) might compromise the model's overall accuracy when tested on large-scale images. It is often an easier problem to directly learn a conditional mapping between the input and label as opposed to learning a full generative model as the S-NCN does [29]. Future work should explore adapting the S-NCN to only learn a conditional mapping as opposed to a full joint distribution over inputs and labels as well as potential mechanisms for pre-training the generative side of the system (which would allow freezing of the synaptic weights for generation and only require updating discriminative parameters – this could potentially reduce the value of $K$ even for more complex sensory inputs). Another drawback, yet also simultaneously a strength, is the fact that the S-NCN is attempting to optimize (online) total discrepancy as opposed to a single, global surface loss. While total discrepancy is one important key to breaking free of backprop and its limitations it also creates a more challenging optimization problem in general, i.e., the neural system must now not only match the values created by data but also ensure its internal activities and its local predictions of each

of them are aligned. While the overall complementary system mitigates catastrophic interference (or the neural cross-talk that would trigger the loss/deletion of previously acquired knowledge), this primarily affects measurements of backward transfer (BWT) but could potentially damage the model's per-task performance, i.e., the main diagonal of its task matrix. Since we do not impose any strong distributional assumptions over the latent activities (such as a clean Gaussian prior as is often done in variational autoencoders), if the S-NCN's estimated value of the latent activities are far from the true posterior, then the S-NCN might produce sub-optimal performance, especially for complex problems. Even though all continual learning systems suffer from this issue (especially most modern-day continual learning ANNs), our model's dual optimization nature could experience this problem more frequently. We believe that integrating memory-aware retrieval from synapses, a mechanism guided by (a brain-like form of) replay, could help direct the system to be closer to the true posterior by avoiding bad local minima when learning continuously.

Additionally, with respect to our proposed task selection mechanism (the functional neural basal ganglia circuit), one notable drawback is that a small refractory period is imposed in order to ensure that enough data is accumulated from the stream to update the competitive task selector's weights. This would be an issue for task streams that constantly introduce tasks with fewer samples than the pre-set refractory period. A subject of our future work is to improve the power /adaptability of the task selection model in the face of more volatile task sequence streams. Another limitation is that the S-NCN is, in effect, a dynamically-expanding architecture: there is an overhead for the task-context memory – one new task context vector would need to be generated/grown for each new (disjoint) task is encountered. While this required parameter growth/generation is not as high as other dynamically-expanding architecture approaches (such as progressive networks [30] or dynamically-expanding networks [31], where many new parameters must be created per task). While the inclusion of relatively few, additional context vector parameters is more desirable, requiring the growing out of parameters at a rate far less than methods such as [30, 31]. To mitigate the cost that even the S-NCN imposes, we remark that the S-NCN's generative cortex could be adapted to induce its own form of layer-wise replay as one alternative, similar to [5], or that another circuit could be designed to potentially learn how to compress these task contexts by reducing redundancy exploiting overlap/redundancy between contexts (serving as a form of efficient long-term memory).

Finally, a more obvious drawback is that the S-NCN's error synapses also increase the memory footprint of the overall model. It would be advisable, when using/applying a model like the S-NCN on other continual learning problems, to select the number of hidden layers and number of neurons in each layer based on model capacity, i.e., compute the total number of (generative and error) synapses that would result from making the neural structure more complex or deeper. A more long-term, promising means of mitigating the increased memory footprint would be to design error units further inspired by actual neurons – instead of assuming a one-to-one mapping (one error neuron per state neuron), design small pools of neurons that are responsible for computing the mismatch activities for large groups of state neurons. This is a key solution to investigate in future work.

## Creating Task Orderings #1 and #2

To create our sequential learning benchmarks, we utilize the MNIST, Fashion MNIST, and Google Draw datasets to create various sets of "subtasks", or rather, classification problems that involve different classes of the original set of each full dataset. In this paper, we create a 6 task sequence, $\{T_1, T_2, T_3, T_4, T_5, T_6\}$, from these datasets, where two tasks are generated from each specific dataset. To create the task splits, we create data subsets based on minimizing the amount of knowledge transfer across data splits, specifically by examining the amount of stroke overlap in the images across classes, yielding a challenging problem. For equal number of classes, the splits we created were: MNIST set #1, $M1 = \{0, 8, 3, 5, 2\}$, MNIST set #2, $M2 = \{1, 4, 6, 7, 9\}$, Fashion MNIST set #1, $FM1 = \{$top, trouser, pullover, dress, coat$\}$, Fashion MNIST set #2, $FM2 = \{$sandal, shirt, sneaker, bag, ankle boot$\}$, Google Draw set #1, $GD1 = \{$objects that were car or bike variants $\}$, and Google Draw set #2, $GD2 = \{$objects belong to variants of airplanes or submarines $\}$. For a task sequence, we create two scenarios: 1) where number of classes are equal for all tasks (i.e., 5 classes in our setup), and 2) where number of classes are unequal (number of classes per task was chosen randomly, omitting the number 5 as an option). In our experiments, we investigate two task orderings (Ordering #1 and Ordering #2). We compute the color index similarity [32] between every pair of tasks (as a proxy for task similarity) and randomly chose Orderings # 1 and # 2 so that the color similarity between

adjacent tasks was higher for Ordering #1 ("High Color Sim." for high color similarity) than for Ordering #2 ("Low Color Sim." for low color similarity), hence task transfer should be easier for Ordering #1 than #2. These particular orderings could be considered to be "harder" and "easier" task orderings, respectively, since it is possible that a the difference in color-index would make it easier to differentiate the tasks (the diversity of inputs from the first few tasks might even improve the performance, as it would be in the case of Ordering # 2). We can see this reflected by the fact that all models (the S-NCN and the baselines) perform a bit better in general on Ordering # 2 (low color similarity or "easier" ordering) than on Ordering # 1 (high color similarity or "harder" ordering).

## Expanded Results for Task Orderings # 1 and # 2

**Metrics for Quantifying Memory Retention:** The formulas for ACC and BWT are:

$$\text{ACC} = \frac{1}{T} \sum_{i=1}^{T} R_{T,i}, \quad \text{BWT} = \frac{1}{T-1} \sum_{i=1}^{T-1} R_{T,i} - R_{i,i}. \tag{13}$$

In addition, **we propose two additional, complementary metrics**, with the motivation that these metrics examine aspects of forgetting and capacity not clearly captured by ACC or BWT. Our two measures, True BWT (TBWT) and Cumulative BWT for task $T_t$ (CBWT(t)), are defined as follows:

$$\text{CBWT(t)} = \frac{1}{T-t} \sum_{i=t+1}^{T} R_{i,t} - R_{t,t} \tag{14}$$

$$\text{TBWT} = \frac{1}{T-1} \sum_{i=1}^{T-1} R_{T,i} - G_{i,i} \tag{15}$$

where $G_{i,i}$ is the performance of an independent classifier trained on task $i$ (in our experiments, this was a full capacity MLP trained via backprop). TBWT relates the degradation in prior task performance by replacing the diagonal of task matrix $R$ with a "gold standard", which is the performance of a model that, in isolation, is able to allocate its full capacity to a particular task. CBWT(t) is a task-specific metric, where we instead examine a particular column of $R$, and measure the total amount of forgetting throughout the sequential learning process, instead of simply examining the final performance at the end (bottom row of $R$) as BWT can only do. CBWT(t) would punish models that suffer large dips in performance in the middle of learning (but not necessarily at the end), and would be better suited for characterizing forgetting in stream settings than BWT.

**Discussion:** Results are reported in Tables 3 and 4 (an expanded version of the original one in the main paper). Each simulation was run 10 times, each trial using a unique seed for pseudo-random number generation, we report both mean and standard deviation. As we observe in our experimental results, we see that all of our S-NCN models exhibit improved memory retention over simple baselines, such as backprop, and more notably, EWC. However, we see that incorporating task-driven lateral inhibition in facilitating gradual forgetting as opposed to catastrophic forgetting, as evidenced by the very competitive performance of both Lat1-S-NCN and Lat2-S-NCNs, with Lat1-S-NCN outperforming all baselines consistently, in terms of both ACC and BWT. This result is robust across both task sequences and equal/ unequal class settings. It is further important to note that the meta-parameter settings used for the various S-NCNs were only tweaked minorly with the same values across all of the settings/scenarios. The observation that lateral inhibition improves the neural computation of our interactive network further corroborates the result of [33], though it focused on models trained via contrastive Hebbian learning.

Upon examination of Table 3, in terms of TBWT and CBWT(1)[1], the proposed lateral S-NCNs still outperform the baselines. The key difference is that we see that the lateral S-NCNs actually do retain prior information throughout learning and do not simply just recover it at the end.

---

[1]We measure CBWT for task $T_1$, since this measures total forgetting over the full length of the task sequence.

Table 3: Alternative metrics reported for task sequence orderings #1 and #2 (higher values are better).

| | Ordering #1 (High Color Sim.) | | | | Ordering #2 (Low Color Sim.) | | | |
| | Equal | | Unequal | | Equal | | Unequal | |
| | TBWT | CBWT | TBWT | CBWT | TBWT | CBWT | TBWT | CBWT |
|---|---|---|---|---|---|---|---|---|
| Backprop | -0.426 | -0.358 | -0.547 | -0.496 | -0.422 | -0.566 | -0.602 | -0.639 |
| EWC | -0.409 | -0.332 | -0.516 | 0.477 | -0.400 | -0.521 | -0.599 | -0.611 |
| Md-IMM | -0.388 | -0.296 | -0.481 | -0.390 | -0.355 | -0.411 | -0.521 | -0.429 |
| DT+Md-IMM | -0.342 | -0.281 | -0.466 | -0.355 | -0.340 | -0.399 | -0.491 | -0.389 |
| L2+DT+Md-IMM | -0.301 | -0.250 | -0.422 | -0.318 | -0.281 | -0.355 | -0.401 | -0.378 |
| HAT | -0.277 | -0.252 | -0.341 | -0.291 | -0.200 | -0.341 | -0.285 | -0.328 |
| S-NCN | -0.397 | -0.509 | -0.507 | -0.623 | -0.373 | -0.252 | -0.564 | -0.396 |
| Lat1-S-NCN | **-0.048** | **-0.074** | **-0.081** | **-0.048** | **-0.032** | **-0.150** | **-0.086** | **-0.037** |
| Lat2-S-NCN | -0.226 | -0.304 | -0.270 | -0.310 | -0.145 | -0.215 | -0.198 | -0.273 |

Table 4: Generalization metrics (10 trials) for sequence orderings # 1 & #2 (higher values are better).

| | Ordering #1: $\{M1, M2, GD1, FM1, FM2, GD2\}$ **(High Color Sim.)** | | | |
| | **Equal** | | **Unequal** | |
| | **ACC** | **BWT** | **ACC** | **BWT** |
| Backprop | $0.241 \pm 0.050$ | $-0.759 \pm 0.030$ | $0.185 \pm 0.048$ | $-0.791 \pm 0.048$ |
| Backprop+DO | $0.251 \pm 0.049$ | $-0.711 \pm 0.030$ | $0.178 \pm 0.049$ | $-0.733 \pm 0.049$ |
| EWC | $0.280 \pm 0.023$ | $-0.714 \pm 0.030$ | $0.185 \pm 0.046$ | $-0.726 \pm 0.039$ |
| EWC+DO | $0.231 \pm 0.029$ | $-0.687 \pm 0.029$ | $0.184 \pm 0.044$ | $-0.710 \pm 0.038$ |
| Mean-IMM | $0.279 \pm 0.019$ | $-0.465 \pm 0.024$ | $0.210 \pm 0.041$ | $-0.499 \pm 0.043$ |
| Md-IMM | $0.521 \pm 0.027$ | $-0.392 \pm 0.023$ | $0.480 \pm 0.039$ | $-0.240 \pm 0.040$ |
| DT+Mean-IMM | $0.321 \pm 0.023$ | $-0.430 \pm 0.020$ | $0.300 \pm 0.044$ | $-0.471 \pm 0.044$ |
| DT+Md-IMM | $0.530 \pm 0.024$ | $-0.387 \pm 0.021$ | $0.551 \pm 0.042$ | $-0.220 \pm 0.042$ |
| L2+Mean-IMM | $0.301 \pm 0.022$ | $-0.443 \pm 0.022$ | $0.250 \pm 0.038$ | $-0.492 \pm 0.046$ |
| L2+Md-IMM | $0.491 \pm 0.020$ | $-0.376 \pm 0.023$ | $0.480 \pm 0.039$ | $-0.235 \pm 0.041$ |
| L2+DT+Mean-IMM | $0.354 \pm 0.029$ | $-0.390 \pm 0.021$ | $0.351 \pm 0.039$ | $-0.421 \pm 0.046$ |
| L2+DT+Md-IMM | $0.532 \pm 0.025$ | $-0.237 \pm 0.027$ | $0.520 \pm 0.040$ | $-0.240 \pm 0.045$ |
| HAT | $0.550 \pm 0.019$ | $-0.211 \pm 0.020$ | $0.492 \pm 0.031$ | $-0.231 \pm 0.036$ |
| S-NCN (ours) | $0.421 \pm 0.022$ | $-0.408 \pm 0.026$ | $0.352 \pm 0.016$ | $-0.476 \pm 0.020$ |
| S-NCN-relu (ours) | $0.398 \pm 0.009$ | $-0.430 \pm 0.012$ | $0.352 \pm 0.008$ | $-0.470 \pm 0.011$ |
| Lat1-S-NCN (ours) | $\mathbf{0.716 \pm 0.013}$ | $\mathbf{-0.031 \pm 0.017}$ | $\mathbf{0.713 \pm 0.011}$ | $\mathbf{-0.041 \pm 0.012}$ |
| Lat2-S-NCN (ours) | $0.573 \pm 0.020$ | $-0.236 \pm 0.0258$ | $0.554 \pm 0.038$ | $-0.235 \pm 0.042$ |

| | Ordering #2: $\{GD2, M1, FM2, M2, GD1, FM1\}$ **(Low Color Sim.)** | | | |
| | **ACC** | **BWT** | **ACC** | **BWT** |
| Backprop | $0.303 \pm 0.030$ | $-0.644 \pm 0.037$ | $0.287 \pm 0.043$ | $-0.671 \pm 0.043$ |
| Backprop+DO | $0.285 \pm 0.032$ | $-0.587 \pm 0.031$ | $0.266 \pm 0.042$ | $-0.610 \pm 0.044$ |
| EWC | $0.303 \pm 0.031$ | $-0.643 \pm 0.033$ | $0.291 \pm 0.039$ | $-0.663 \pm 0.047$ |
| EWC+DO | $0.302 \pm 0.033$ | $-0.558 \pm 0.032$ | $0.281 \pm 0.039$ | $-0.586 \pm 0.046$ |
| Mean-IMM | $0.453 \pm 0.026$ | $-0.170 \pm 0.031$ | $0.402 \pm 0.036$ | $-0.274 \pm 0.035$ |
| Md-IMM | $0.584 \pm 0.027$ | $-0.091 \pm 0.030$ | $0.533 \pm 0.034$ | $-0.230 \pm 0.036$ |
| DT+Mean-IMM | $0.558 \pm 0.021$ | $-0.128 \pm 0.029$ | $0.510 \pm 0.033$ | $-0.254 \pm 0.035$ |
| DT+Md-IMM | $0.591 \pm 0.020$ | $-0.088 \pm 0.032$ | $0.528 \pm 0.036$ | $-0.211 \pm 0.039$ |
| L2+Mean-IMM | $0.465 \pm 0.021$ | $-0.156 \pm 0.033$ | $0.430 \pm 0.039$ | $-0.271 \pm 0.032$ |
| L2+Md-IMM | $0.576 \pm 0.028$ | $-0.99 \pm 0.038$ | $0.511 \pm 0.036$ | $-0.266 \pm 0.039$ |
| L2+DT+Mean-IMM | $0.587 \pm 0.025$ | $-0.105 \pm 0.033$ | $0.528 \pm 0.038$ | $-0.253 \pm 0.043$ |
| L2+DT+Md-IMM | $0.630 \pm 0.029$ | $-0.076 \pm 0.030$ | $0.551 \pm 0.037$ | $-0.201 \pm 0.041$ |
| HAT | $0.596 \pm 0.026$ | $-0.114 \pm 0.029$ | $0.563 \pm 0.031$ | $-0.210 \pm 0.044$ |
| S-NCN (ours) | $0.444 \pm 0.017$ | $-0.393 \pm 0.0210$ | $0.272 \pm 0.013$ | $-0.587 \pm 0.014$ |
| S-NCN-relu (ours) | $0.431 \pm 0.009$ | $-0.398 \pm 0.010$ | $0.286 \pm 0.007$ | $-0.559 \pm 0.008$ |
| Lat1-S-NCN (ours) | $\mathbf{0.721 \pm 0.014}$ | $\mathbf{-0.042 \pm 0.013}$ | $\mathbf{0.667 \pm 0.011}$ | $\mathbf{-0.097 \pm 0.013}$ |
| Lat2-S-NCN (ours) | $0.633 \pm 0.028$ | $-0.170 \pm 0.033$ | $0.5778 \pm 0.035$ | $-0.211 \pm 0.042$ |

## Expanded Benchmark Results

To start, we describe the three forms of the lateral competition we designed for the S-NCN. They were specifically as follows:

1. $f^\ell(\mathbf{z}^\ell, \mathbf{g}_t^\ell) = \mathbf{I} \cdot \mathbf{z}^\ell$, which means that the lateral inhibitory matrix is fixed to a diagonal matrix $\mathbf{I}$ and forces the model to ignore the task embedding (in the Appendix, we denote this as "NoLat-S-NCN"),

2. $f^\ell(\mathbf{z}^\ell, \mathbf{g}_t^\ell) = \left(\mathbf{I} \odot \mathbf{V}^\ell\right) \cdot \mathbf{z}^\ell$, where the matrix $\mathbf{V}^\ell = \text{BKWTA}(\mathbf{g}_t^\ell, K) \cdot \text{BKWTA}((\mathbf{g}_t^\ell)^T, K)$ and $\text{BKWTA}(\mathbf{v}, K)$ is the binarized $K$ winners-take-all function, yielding a binary vector with 1 at the index of each of the $K$ winning units (in the Appendix, we denote this as "Lat1-S-NCN"),

3. $f^\ell(\mathbf{z}^\ell, \mathbf{g}_t^\ell) = \max\left(0, \mathbf{z}^\ell - \left((\mathbf{V}^\ell \odot (\mathbf{g}_t^\ell \cdot (\mathbf{g}_t^\ell)^T)) \cdot \mathbf{z}^\ell\right)\right)$, where: $\mathbf{V}_{i,j}^\ell = \{\alpha, \text{if } i \neq j, \text{else}, 0\}$ (in the Appendix, we denote this as "Lat2-S-NCN").

In the last two forms of the competition function, we see that lateral inhibition is a function of an evolving context vector $\mathbf{g}^\ell$, triggered by the presence of the task signal $t_i$. The above competition functions correspond to different designs of lateral suppression patterns: (1) corresponds to no lateral suppression, (2) corresponds to shutting off neurons that are not task-relevant driven by a task selector, (3) corresponds to a task-driven, real-valued lateral matrix that scales neural activities.

Table 5: Generalization metrics (10 trials) for Split MNIST, Split Fashion MNIST (FMNIST) and Not-MNIST benchmarks. Note for IMM, we employ the best performing variant, *L2+DT+Md-IMM*. Above dashed line are single-head models & below are multi-head models (except the S-NCN).

| | MNIST | | FMNIST | | NotMNIST | |
| --- | --- | --- | --- | --- | --- | --- |
| | ACC | BWT | ACC | BWT | ACC | BWT |
| EWC | $0.760 \pm 0.030$ | $-0.210 \pm 0.011$ | $0.739 \pm 0.020$ | $-0.201 \pm 0.011$ | $0.790 \pm 0.020$ | $-0.176 \pm 0.010$ |
| VCL | $0.980 \pm 0.210$ | $-0.003 \pm 0.002$ | $0.980 \pm 0.20$ | $-0.002 \pm 0.003$ | $0.953 \pm 0.003$ | $-0.004 \pm 0.002$ |
| IMM | $0.951 \pm 0.018$ | $-0.007 \pm 0.003$ | $0.950 \pm 0.013$ | $-0.005 \pm 0.003$ | $0.925 \pm 0.011$ | $-0.006 \pm 0.002$ |
| HAT | $0.972 \pm 0.011$ | $-0.040 \pm 0.002$ | $0.968 \pm 0.011$ | $-0.004 \pm 0.002$ | $0.942 \pm 0.009$ | $-0.005 \pm 0.002$ |
| GEM | $0.922 \pm 0.110$ | $\mathbf{+0.001 \pm 0.002}$ | $0.930 \pm 0.12$ | $+0.001 \pm 0.003$ | $0.919 \pm 0.021$ | $-0.003 \pm 0.002$ |
| DGR | $0.911 \pm 0.300$ | $-0.011 \pm 0.002$ | $0.915 \pm 0.25$ | $-0.013 \pm 0.001$ | $0.920 \pm 0.015$ | $-0.014 \pm 0.003$ |
| Rtf | $0.925 \pm 0.200$ | $-0.009 \pm 0.003$ | $0.930 \pm 0.25$ | $-0.009 \pm 0.005$ | $0.922 \pm 0.012$ | $-0.011 \pm 0.004$ |
| EWC | $0.190 \pm 0.030$ | $-0.357 \pm 0.015$ | $0.199 \pm 0.06$ | $-0.350 \pm 0.012$ | $0.186 \pm 0.020$ | $-0.361 \pm 0.010$ |
| NR+M1 | $0.906 \pm 0.870$ | $-0.050 \pm 0.001$ | $0.900 \pm 0.810$ | $-0.060 \pm 0.003$ | $0.890 \pm 0.030$ | $-0.071 \pm 0.004$ |
| NR+M2 | $0.950 \pm 0.470$ | $-0.100 \pm 0.003$ | $0.948 \pm 0.380$ | $-0.090 \pm 0.003$ | $0.880 \pm 0.028$ | $-0.103 \pm 0.002$ |
| SI | $0.197 \pm 0.110$ | $-0.367 \pm 0.014$ | $0.198 \pm 0.10$ | $-0.370 \pm 0.013$ | $0.161 \pm 0.030$ | $-0.370 \pm 0.010$ |
| MAS | $0.195 \pm 0.290$ | $-0.340 \pm 0.010$ | $0.180 \pm 0.25$ | $-0.340 \pm 0.010$ | $0.178 \pm 0.060$ | $-0.341 \pm 0.011$ |
| Lwf | $0.846 \pm 0.340$ | $-0.120 \pm 0.001$ | $0.875 \pm 0.30$ | $-0.130 \pm 0.003$ | $0.626 \pm 0.091$ | $-0.130 \pm 0.004$ |
| ICarl | $0.940 \pm 0.410$ | $-0.100 \pm 0.004$ | $0.960 \pm 0.40$ | $-0.110 \pm 0.005$ | $0.887 \pm 0.102$ | $-0.109 \pm 0.007$ |
| Lucir | $0.940 \pm 0.310$ | $-0.103 \pm 0.007$ | $0.950 \pm 0.34$ | $-0.110 \pm 0.005$ | $0.935 \pm 0.093$ | $-0.101 \pm 0.006$ |
| Bic | $0.901 \pm 0.860$ | $-0.139 \pm 0.009$ | $0.890 \pm 0.85$ | $-0.160 \pm 0.009$ | $0.851 \pm 0.099$ | $-0.155 \pm 0.009$ |
| Mnem | $0.960 \pm 0.320$ | $-0.991 \pm 0.005$ | $0.968 \pm 0.30$ | $\mathbf{+0.007 \pm 0.006}$ | $0.950 \pm 0.071$ | $-0.011 \pm 0.007$ |
| S-NCN | $\mathbf{0.981 \pm 0.300}$ | $-0.005 \pm 0.004$ | $\mathbf{0.982 \pm 0.400}$ | $-0.003 \pm 0.007$ | $\mathbf{0.957 \pm 0.400}$ | $\mathbf{-0.004 \pm 0.005}$ |

On the custom benchmarks (including both orderings #1 and #2), we evaluate four variations of the S-NCN: 1) an S-NCN, with hyperbolic tangent activations and no context-dependent lateral inhibition (S-NCN), 2) an S-NCN with sparsity created by the use of a linear rectifier activation function and no lateral inhibition (S-NCN-relu), 3) an S-NCN with the second variant of our proposed lateral inhibition (Lat1-S-NCN), and 4) an S-NCN with the third variant of our proposed lateral inhibition (Lat2-S-NCN) (All variants used: $\beta = 0.05$, $K = 10$, $\eta_g = 0.9$, $\eta_e = 0.01$, $\alpha = 0.98$). The last two S-NCN models ("Lat1-S-NCN" and "Lat2-S-NCN") were driven by the FNBG model that we described in Section 3.3 ("The Neural Task Selection Model").

We compare our S-NCN model (specifically, our best-performing one from the experiment in the last section – the Lat1-S-NCN) to the following approaches that have been proposed over the years to combat catastrophic interference: Naive rehearsal with memory (NR+M1 & NR+M2), EWC, synaptic intelligence (SI), MAS [18], Lwf [19], GEM [20], DGR [34], Rtf [35], ICarl [21], Lucir [22], Bic [36], and Mnemonics [23]. In Table 5, we report model ACC and BWT, averaged over 10 trials, offering not only an extensive and comprehensive comparison of competitive methods, but also demonstrating that, for all three benchmarks, **our proposed S-NCN outperforms all of them**, demonstrating the power afforded by challenging the very assumptions underlying modern-day artificial neural systems and designing models with stronger grounding in neuro-biology. Furthermore, it is astounding to see that the S-NCN outperforms/matches performance with not only the single-head models but also with the multi-head models (except GEM), which enjoy an easier version of the problem since they are permitted to utilize a different classifier per task. Finally, it is critical to note that **our proposed S-NCN is a complementary neural system that learns without explicitly-provided task descriptors**, i.e., in other words, the model learns to compose its own task contexts in a data-dependent manner.

Note that, in the online setting, split FashionMNIST appears to be simpler, given that the S-NCN readily learns to generate rough forms of bigger objects (shirt/shoes/pants) and associate these with labels early whereas digits/characters are a bit more intricate and take longer to learn.

## S-NCN Task Accuracy Curves

See Figure 1 (below) for a visual depiction of the S-NCN's task accuracy over time across tasks for all three continual learning benchmarks investigated in this paper.

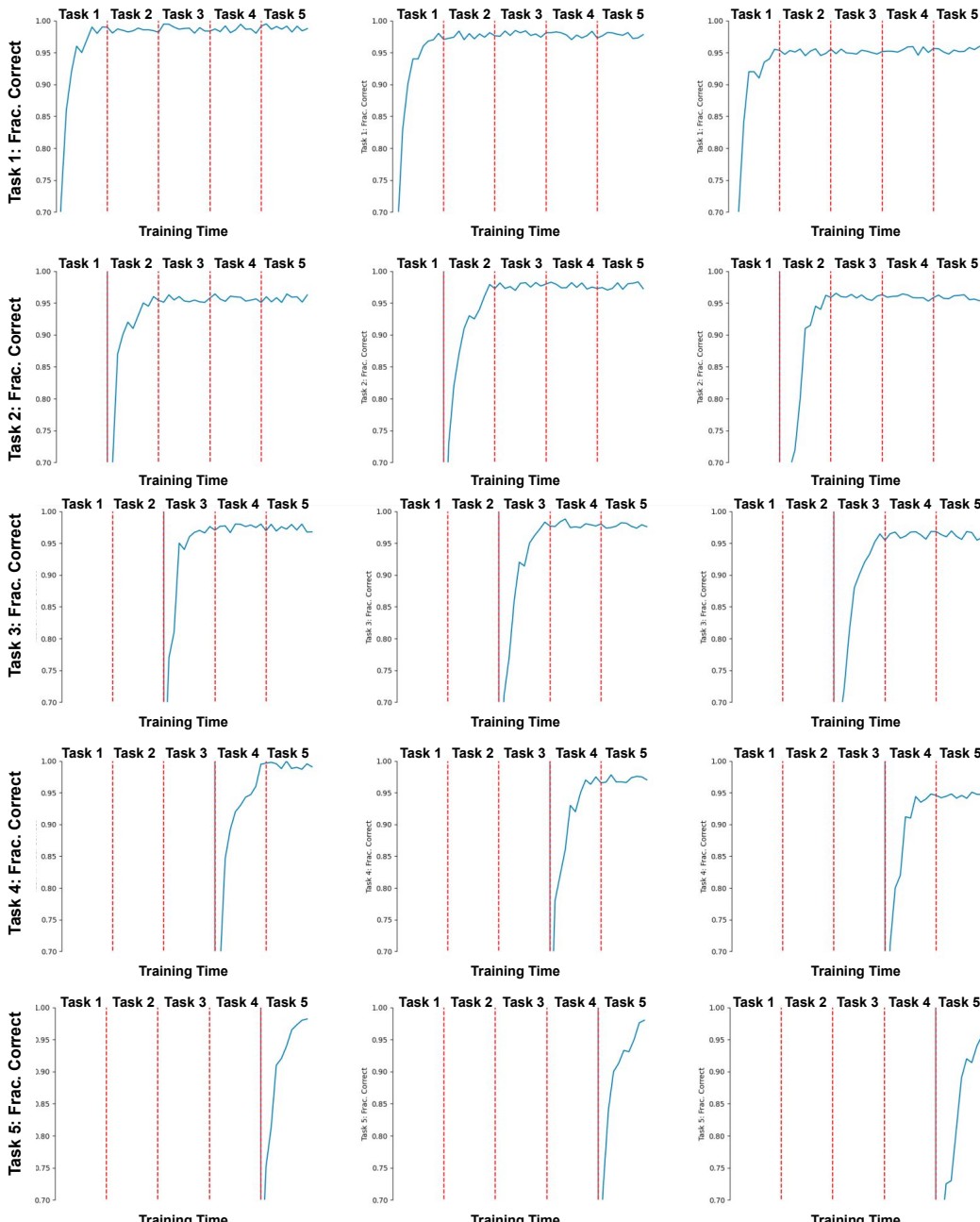

Figure 1: Task accuracy curves for the S-NCN measured across the five tasks within the (Left column) Split MNIST, (Middle column) Split Fashion MNIST, and (Right column) Split NotMNIST benchmarks (red vertical lines indicate actual task boundaries). Y-axis depicts the fraction correct while the x-axis depicts (online) training iteration (or one single epoch/pass through each task). Each row, as indicated by the Y-axis label, represents the perspective of a different task, e.g., row 1 corresponds to performance on Task 1 as the S-NCN learns across all tasks, row 2 corresponds to performance on Task 2 as the S-NCN learns across all tasks, etc.