# OpenReview forum: "Lifelong Neural Predictive Coding: Learning Cumulatively Online without Forgetting"
_NeurIPS.cc/2022/Conference — NeurIPS 2022 Accept_

### Official Review · Reviewer_qc3j · 2022-07-05

**Rating:** 7
**Confidence:** 3
**Soundness:** 3 good
**Presentation:** 3 good
**Contribution:** 3 good

**Summary:**

This work uses a generative predictive coding model to perform continuous learning. The goal of such models is ti perform learning and inference on multiple tasks by using the same network, with the hope that (1) training on a new dataset won't make the network forget previously learned information, and (2) that the network will actually use previously learned information to perform better inference on the new task. To this end, the authors propose a generative PC model, conceptually similar to the one proposed by Rao and Ballard, with multiple differences:

1) Feedforward and feedback weights are different;

2) Multiple regularizers are used to improve the performance (modulation factors);

3) To make continual learning possible, learnable task embedding vectors, that depend on specific tasks and are separated using a contrastive loss, are added.

The method is then tested on multiple benchmarks, with good results. As a disclaimer, I am not an expert on continual learning, hence I'm judging the experiments based on the reported results. After a couple of google searches, it seems to me that the authors have well picked the baselines and datasets, but I may be wrong here.

**Questions:**

Above, I have highlighted the three differences between generative PCNs and your proposed method above: Feedback connections, task embedding vectors, and regularisers. Of course, task embedding vectors are essential for continual learning, and are probably the main reason your model works well.  However, why have you decided to implement the other two differences? so, more in detail:

1) How would a model with symmetric connections behave?

2) How important is the regularization in the final performance?

I'm asking this, as I have the feeling that removing one of the two, would harm performance, but I'm having troubles understanding why. Hence, I would like to read a discussion that better tackles this. Even better, I would like to see experiments on one task made by using a network where  (1) and/or (2) are removed, and how the performance compare with the final model.


Minor:

Equation 5 is not that clear to me, what is M?

WTA seems to only depend on the value and not on the absolute value, right? Why this choice?

Feedback weights are called "recurrent weights". As a reader, I often associate "recurrent" with "lateral connections", and not feedback ones. Maybe changing the name would make this clear.

Equation 4: shouldn't it be g^l_{t+1} instead of g_l_t.



**Limitations:**

Well and largely discussed in the supplementary material.

**Strengths And Weaknesses:**

Pros:

The idea of the method is nice, and the experimental results confirm the intuition of the authors. It is original, as I'm not aware of any application of generative predictive coding networks (PCNs) to continual learning. The paper reads ok: it is well written, but sometimes the large number of details makes the text flow heavy. To conclude, this work could be significant to two different communities: computational neuroscience, more in detail the branch that tries to tackle machine learning problems this way, and continual learning.

Cons:

While being a generative PCN, the literature about these models is not discussed properly: although some of the relevant work is actually cited In block in line 197, a proper discussion feels needed. Particularly, a proper comparison with the models that inspired your architecture would raise multiple interesting questions that would be nice to discuss in the paper, that I will state in the next paragraph.

---

> ### Author Response · Authors · 2022-08-02
> **Response to Reviewer qc3j**
>
> We would like to thank Reviewer qc3j for their comments and questions.
>
> **While being a generative PCN, the literature about these models is not discussed properly: although some of the relevant work is actually cited In block in line 197, a proper discussion feels needed. Particularly, a proper comparison with the models that inspired your architecture...I will state in the next paragraph.**
>
> We will include a more detailed discussion of the models that inspired this architecture in the paper/appendix. We also want to point out that in the original submission we did write a sub-section in the Appendix/Supplementary Material titled "Design Motivation/Intuition" which includes discussion of elements that inspired us. However, we will expand it given the questions raised by Reviewer qc3j.
>
> Questions/Answers:
>
> **Above, I have highlighted the three differences between generative PCNs and your proposed method above: Feedback connections, task embedding vectors, and regularisers. Of course, task embedding vectors are essential for continual learning, and are probably the main reason your model works well. However, why have you decided to implement the other two differences? so, more in detail:**
>
> **How would a model with symmetric connections behave?**
>
> A model without separate feedback connections (i.e., symmetric connections) would behave quite similarly and favorably offer a reduction in memory cost (one does not need to store separate feedback/error matrices in memory) – one could certainly swap out E^l with (W^l)^T if desired. However, by utilizing separate learnable feedback synapses, we resolve the weight transport problem (a biological criticism of backprop where error/teaching information is carried backwards along the same synapses that were used to forward propagate information) in the interest of constructing a more neurobiologically faithful circuit. Interestingly enough, we found that using learnable feedback synapses improves the generative modeling/reconstruction ability of the generative cortex (particularly in the online case) even though we did not investigate/highlight this in this work and this change in generative performance did not really seem to impact the classification accuracy (arguably, discrimination is easier than generation).
> We will include a discussion of this in the revised appendix.
>
> **How important is the regularization in the final performance?**
>
> The modulation matrices induce a form of synaptic scaling, which are not intended to relate to fighting forgetting (though see response to Reviewer 36Ec for a potential extension where they could). Their purpose is to improve the stability of the learning process given that our particular form of predictive coding would require post-update weight normalization (as in the case of sparse coding – this is harder to argue is bio-plausible) whereas the modulation matrices (which has evidence that suggests bio-plausibility via synaptic scaling). If one omitted the modulation terms and normalized the synapses every update, the performance would largely be the same.
>
> ***Minor Points:***
>
> **Equation 5 is not that clear to me, what is M?**
>
> The learnable projection matrix M^l maps the task pointer produced by the basal ganglia to the task contexts for a given layer. Also, please see the sub-section “Context Units” (line 103) earlier in the paper where M^l was first mentioned/introduced. We have also introduced an additional symbol \mathcal{M} = \{M^1, …, M^L\} (which houses all the memory matrices) in that section as well as in Algorithms 1-2 (this specific update has now been made to the current OpenReview draft).
>
> **WTA seems to only depend on the value and not on the absolute value, right? Why this choice?**
>
> Yes, the BKWTA (binary K-WTA function we designed) depends on the value not absolute value. This is b/c the update rule for the task contexts (Eqn 5) can result in negative numbers in the context – although, if one desired to only have non-negative numbers in g^l, one could first apply a linear rectifier max(0, v) to ensure only positive numbers – absolute value would not be useful in this context as the very negative numbers in our g^l indicate neural units that the S-NCN determines should not be used / play a role in the current task.
>
> **Feedback weights are called "recurrent weights". As a reader, I often associate "recurrent" with "lateral connections", and not feedback ones. Maybe changing the name would make this clear.**
>
> We have omitted the use of recurrent and now use “feedback weights” as suggested in the revised paper (this specific update has now been made to the current OpenReview draft) .
>
> **Equation 4: shouldn't it be g^l_{t+1} instead of g_l_t.**
>
> Yes, this is a typo and we have corrected it in the revised paper (this specific update has now been made to the current OpenReview draft).

---

> > ### Comment · Reviewer_qc3j · 2022-08-06
> > **Answer to the authors**
> >
> > I thank the authors for the detailed answer and updates. I have raised my score to an acceptance.

---

> > > ### Author Response · Authors · 2022-08-08
> > > **Response to Reviewer qc3j Response: Thank you!**
> > >
> > > We would like to thank Reviewer qc3j for their score raise, and again, for their helpful comments/questions!

---

### Official Review · Reviewer_WKzA · 2022-07-11

**Rating:** 5
**Confidence:** 3
**Soundness:** 3 good
**Presentation:** 3 good
**Contribution:** 2 fair

**Summary:**

The authors propose a biologically inspired/plausible neural network for continual learning. The proposed method leverages a context mechanism to learn multiple tasks while minimising forgetting; and compare favourably against baselines across standard benchmark tasks.

**Questions:**

1 - It seems to me that the main difference between the proposed approach and "Learning to Adapt by Minimizing Discrepancy" is the addition of contexts. Could the authors discuss the similarities and the differences?

2 - Why do all the baselines use same learning rate (\lambda = 0.01)? This seems quite high for conventional neural networks used in conjunction with conventional preprocessing.

3 - The proposed model has two independent learning components: "predictive" weights and contexts. Such models, that aren't trained end-to-end, can be challenging to train due to hyperparameter sensitivity. Can the authors comment on this?

**Limitations:**

Limitations are adequately addressed.

**Strengths And Weaknesses:**

The formalism based on predictive coding is attractive because of its biological plausibility as well as its flexibility. At face value, the model demonstrates very impressive empirical results by outperforming backprop-based neural networks -- which is not an easy feat. However, the experiments section is missing some important analysis in my view.

Regarding the experiments, I understand that no meta-parameter optimisation is done for the baseline methods, which makes the comparison a bit unfair, but it is not the end of the world. However, I do not understand why the original learning rates are altered and set to 0.01 for all baselines (see Question 2).

Many continual learning papers display learning curves across multiple tasks (eg, the EWC paper). This is very helpful for identifying the degree of learning speed and forgetting. In addition, it is helpful to compare the learning curves across papers, which gives some form of reassurance that the implementations or hyperparameter choices are correct/decent.

In terms of novelty, I would like to learn more about the differences between the proposed approach,  "Learning to Adapt by Minimizing Discrepancy", and "Continual Learning of Recurrent Neural Networks by Locally Aligning Distributed Representations" (Question 1). In fact, this should ideally be discussed in "Related Work".

The paper is written clearly and is easy to understand overall. It would be helpful to show how the update equations can be derived from Equation 6. Because, it seems to me that just differentiating the loss wrt weights result in backprop without zeroing/stopping some gradients. Also, the line 20 in Algorithm 2 should refer to Equation 4 I believe.

I am open to bumping up my rating if my concerns are adequately addressed.

---

> ### Author Response · Authors · 2022-08-02
> **Response to Reviewer WKzA (Part 1)**
>
> We would like to thank Reviewer WKzA for their comments and questions.
>
> **Regarding the experiments, I understand that no meta-parameter optimisation is done for the baseline methods, which makes the comparison a bit unfair...However, I do not understand why the original learning rates are altered and set to 0.01 for all baselines (see Question 2).**
>
> We actually tuned each baseline in this paper. See Line 311 in the original paper – “For each baseline, we tuned hyper-parameters based on their accuracy on each 312 task’s development set.” However, in the main paper (and elsewhere), we noticed that the text in Lines 281-283 in the submission was unclear/vague (these phrases/sentences were meant to refer to the S-NCN and not the baselines themselves – the appendix would contradict this if it this was not the case given that it states that we tuned each baseline to the development/validation set).
> We have fixed/corrected this error in both the main and appendix texts in the revised paper (this specific update has now been made to the current OpenReview draft) .
>
> **Many continual learning papers display learning curves across multiple tasks (eg, the EWC paper). This is very helpful for identifying the degree of learning speed and forgetting...choices are correct/decent.**
>
> Please find in the revised Appendix/Supplementary Material (this specific update has now been made to the current OpenReview draft) containing the requested task accuracy plots for the Split MNIST, Fashion MNIST, and notMNIST benchmarks. These accuracy plots are at the very end of the current updated Appendix.
>
> **In terms of novelty, I would like to learn more about the differences between the proposed approach, "Learning to Adapt by Minimizing Discrepancy", and "Continual Learning of Recurrent Neural Networks by Locally Aligning Distributed Representations" (Question 1)...**
>
> We will include a discussion in the Appendix highlighting the differences between this paper’s S-NCN and [1, 2]. See the answer to Question 1 below for what context we will integrate this into the text of the paper/appendix.
>
> **The paper is written clearly and is easy to understand overall. It would be helpful to show how the update equations can be derived from Equation 6...Also, the line 20 in Algorithm 2 should refer to Equation 4 I believe.**
>
> Yes, line 20 should refer to Equation 4. We have fixed this in the main paper (this specific update has now been made to the current OpenReview draft). We **have now included** a derivation of the S-NCN’s state and weight updates from Equation 6 in the Appendix/Supplementary Material as per Reviewer WKzA's request. Note that the S-NCN entails updating the actual neural activities themselves (the stateful latent vectors z^1, z^2,…, z^L) as well as the synaptic weight matrices (inside of \Theta). Both updates follow from Equation 6. Given that each layer of our generative circuit predicts the activities of another nearby layer, the resulting model is quite distinctive from a feedforward, backprop-based network (a feedforward network must compute each layer-wise activity in serial whereas the S-NCN's generative circuit computes each stateful layer activity in parallel) -- nevertheless, one can start by taking derivatives of Equation 6 as we do in the updated Appendix to show where the final updates/learning rules come from.

---

> > ### Author Response · Authors · 2022-08-02
> > **Response to Reviewer WKzA (Part 2)**
> >
> > Questions/Answers:
> >
> > **1 - It seems to me that the main difference between the proposed approach and "Learning to Adapt by Minimizing Discrepancy" is the addition of contexts. Could the authors discuss the similarities and the differences?**
> >
> > The model proposed in "Learning to Adapt by Minimizing Discrepancy" is not layer-wise parallelizable and is a different formulation of PC for feedforward networks (which was also trained with multiple epochs over the data) under a (temporally) recurrent formulation focused on processing temporal/time-vary data (such as the bouncing ball dataset). This model would not, as is, be resistant to forgetting like the S-NCN is – note that the S-NCN is a complementary system made up of two circuits, i.e., a generative cortex model and a basal ganglia task selection model. Furthermore, notice in Eqn 2 of [1] that the neural state updates are a function of only the bottom-up feedback (and not mediated by top-down expectations as in Eqn 2 of this work). The resulting model in this paper offers, as mentioned in response to Reviewer 36Ec, the potential for asynchronous parallel implementation given that the layers are decoupled yet governed by globally optimizing Eqn 6.
> > Similarly, for [2], this is another temporal formulation of PC focused again on time-varying data sequences. This model would not be, in principle, be all too resistant to forgetting as the S-NCN system would be given that it has no internal notion of task context – the functional basal ganglia of the S-NCN dynamically determines/produces the task context (parameters) that drive the generative cortex, significantly reducing forgetting over task/dataset sequences. Furthermore, the work of [2] focused on unsupervised sequence modeling and did not investigate any form of discriminative learning (e.g., classification) where forgetting is, in our experience, far faster and stronger to observe. (Note that in [2], although some improved memory retention across the three sequence modeling tasks was observed, forgetting over tasks was still apparent, and any improvement was more of a pleasant side-effect rather than the result of a mechanism specialized for safe-guarding against forgetting).
> >
> > [1] Ororbia, Alexander G., et al. "Learning to adapt by minimizing discrepancy." (2017).
> > [2] Ororbia, Alexander, et al. "Continual learning of recurrent neural networks by locally aligning distributed representations." (2020).
> >
> > **2 - Why do all the baselines use same learning rate (\lambda = 0.01)? This seems quite high for conventional neural networks used in conjunction with conventional preprocessing.**
> >
> > The baselines do not use a learning rate of 0.01. This was an error/ambiguously placed sentence. As noted earlier in this response (and in the Appendix), each baseline was tuned specifically to maximize its performance. We have fixed this mistake in the paper (this specific update has now been made to the current OpenReview draft).
> >
> > **3 - The proposed model has two independent learning components: "predictive" weights and contexts. Such models, that aren't trained end-to-end, can be challenging to train due to hyperparameter sensitivity. Can the authors comment on this?**
> >
> > While we agree that non-end-to-end models can be trickier to train in general, we have found that the S-NCN training overall was not terribly sensitive to most of its hyper-parameters – we did not do end up having to do much tuning of the S-NCN in this work (although the baselines require more extensive tuning) and found that roughly the same values worked for all benchmarks investigated in this study. The task context/memory updates work well with eta_g > 0.5 and eta_e chosen within the range [0.005,0.25] (in Equation 4). The basal ganglia synaptic updates worked generally well with \rho > 0.5. The generative cortex was found to reliably work with the settings reported in this paper, and the only notable difference in performance can be found is lambda (learning rate) and beta (state update factor) are changed drastically – if they are too small (lambda < 0.001 and beta < 0.01) the generative model takes too long to learn well (and does not obtain high enough task accuracies in time given that we focus on online/streaming performance with only one pass/epoch) and if they are too high (lambda > 0.2 and beta > 0.4) the learning process can become unstable (using beta = 0.1 works well in general).
> > We will include a discussion of this in the Appendix of the paper.

---

> > > ### Comment · Reviewer_WKzA · 2022-08-07
> > > **Borderline Accept**
> > >
> > > I thank the authors for the detailed reply. Many of my concerns are adequately addressed. Thus, I revise my recommendation from "Borderline Reject" to "Borderline Accept".
> > >
> > > > Please find in the revised Appendix/Supplementary Material (this specific update has now been made to the current OpenReview draft) containing the requested task accuracy plots for the Split MNIST, Fashion MNIST, and notMNIST benchmarks. These accuracy plots are at the very end of the current updated Appendix.
> > >
> > > It is great to see the learning speed; however, it would have been more informative to show the performance on previous tasks (thus, forgetting).
> > >
> > > >  As noted earlier in this response (and in the Appendix), each baseline was tuned specifically to maximize its performance. We have fixed this mistake in the paper (this specific update has now been made to the current OpenReview draft).
> > >
> > > The hyperparameter choices for the baseline methods are still not available as far as I can see, conflicting with the "Yes" answer to the checklist item 3b. It would be helpful to see the tested & selected hyperparameters.

---

> > > > ### Author Response · Authors · 2022-08-09
> > > > **Response to Reviewer WKzA Response: Task Accuracy Plot is Learning Curve Plot (from EWC paper) & Addition of Baseline Hyperparameters**
> > > >
> > > > We thank Reviewer WKzA for their response and the score raise/increase.
> > > >
> > > > To address your two final comments in your response to the response:
> > > >
> > > > **1) "It is great to see the learning speed; however, it would have been more informative to show the performance on previous tasks (thus, forgetting)."**
> > > >
> > > > To clarify (if we understand the request correctly), our task accuracy plot (Figure 1 in the Appendix) does show that there is no forgetting in previous tasks (we produced an albeit cruder version of the task accuracy plot in the EWC paper, i.e., Kirkpatrick et. al, 2017). This plot which we added in the Appendix is directly meant to fulfill the request for a learning curve plot (as shown in the quote below):
> > > > "Many continual learning papers display learning curves across multiple tasks (eg, the EWC paper). This is very helpful for identifying the degree of learning speed and forgetting. In addition, it is helpful to compare the learning curves across papers, which gives some form of reassurance that the implementations or hyperparameter choices are correct/decent."
> > > >
> > > > Figure 1 at the very end of the Appendix is the requested learning curve/task accuracy plot. However, due to concern that our learning curve plot might not have been clear enough, we have further modified it to make it clearer as the text in the original version was too small/faint. ***Please see the updated Figure 1 in the Appendix to see the revision.***
> > > > This plot, much like the one in the EWC paper, only shows a reasonable range of fraction correct values (between 0.8 to 1.0 similar to the EWC paper's -- since before learning on a given task occurs, all model's generally perform below 60% accuracy). As observed in the clearer Figure 1, the S-NCN does not exhibit much forgetting in previously encountered tasks for all three benchmarks. In Figure 1, the left column shows the Split MNIST curves, the middle column shows the Split Fashion MNIST curves, and the right column shows the Split NotMNIST curves. Each row corresponds to the S-NCN's performance from the perspective of a particular task, e.g., row 1 corresponds to Task 1 performance across all tasks, row 2 corresponds to Task 2 performance across all tasks, etc -- we have modified the caption of Figure 1 to include this explanation to clear things up further.
> > > > (Also please note that the S-NCN is trained online whereas the models in the EWC were trained for many epochs per task.)
> > > >
> > > > ***Note:*** We realize another potential source of confusion might relate to our use of "Training Time" on Figure 1's x-axis, which was chosen to conform to the exact x-axis choice of Kirkpatrick et. al, 2017, who also label that x-axis as "Training Time". However, another way of labeling the x-axis could be "Training Iteration" if this desirable (and we can make this further change if requested/desired).
> > > >
> > > > **2. "It would be helpful to see the tested & selected hyperparameters."**
> > > >
> > > >  We have added the hyper-parameter values chosen (that resulted from our tuning process) for each baseline -- these now appear in the new Table 2 in the Appendix.
> > > >
> > > > Please note that, again, all text changes are highlighted in blue (and that Tables 2 and 3 appear in blue while Figure 1's caption is blue to indicate it is new).

---

### Official Review · Reviewer_36Ec · 2022-07-13

**Rating:** 5
**Confidence:** 5
**Soundness:** 2 fair
**Presentation:** 2 fair
**Contribution:** 3 good

**Summary:**

The authors in the paper propose a complementary system trained using local learning rules and bio-inspired mechanisms inspired from basal ganglia, such as lateral inhibition and gating based on context, task selection using context memory. The authors propose a decomposed learning mechanism which eliminates the need for weight transport mechanisms. The methods proposed in the paper combine multiple mechanisms to demonstrate continual learning capabilities on image recognition datasets.

**Questions:**

1. Based on the algorithm, the initial prediction states are being computed using a top-down approach. So, how is $z^l$ being computed for the initial state, is it a random initialization or is it directly the one hot encoded label? If the z values for the intermediate labels are initialized to 0, how would we compute the $z^l_u$ values for the corresponding layers based on equation 1.
2. "Concretely, our approach could be classified as task incremental learning (Domain-IL) [21] with the exception that no task descriptors are used at both training and test time (an aspect of Class-IL [21])" - This sentence needs to be clarified and rephrased. Task IL is not Domain IL.
3. Isn’t there an overhead for storing the task-specific context memory. And would that not also increase/grow with the number of tasks? there are other replay-based approaches which also operate with very limited and small buffer sizes or there are other generative based models like brain inspired replay, so the point regarding the proposed model being more memory efficient needs to be discussed in the paper.
4. The learning rule decomposing the weight update towards local layers without transmitting the error backwards has been already proposed by other surrogate gradient-based mechanisms. So, however using local knowledge only for learning is novel. How would you compare the proposed approach to the other surrogate gradient-based approaches?
5. The proposed weight update rule uses modulation factors as an optimizer factor for updating the weights. The modulation factors are similar to some of the previously proposed approaches

    * Beaulieu, S., Frati, L., Miconi, T., Lehman, J., Stanley, K. O., Clune, J., & Cheney, N. (2020). Learning to continually learn. arXiv preprint arXiv:2002.09571.

    * Tsuda, B., Tye, K. M., Siegelmann, H. T., & Sejnowski, T. J. (2020). A modeling framework for adaptive lifelong learning with transfer and savings through gating in the prefrontal cortex. Proceedings of the National Academy of Sciences, 117(47), 29872-29882.

    * Imam, N., & Cleland, T. A. (2020). Rapid online learning and robust recall in a neuromorphic olfactory circuit. Nature Machine Intelligence, 2(3), 181-191.

    * Madireddy, S., Yanguas-Gil, A., & Balaprakash, P. (2020). Neuromodulated neural architectures with local error signals for memory-constrained online continual learning. arXiv preprint arXiv:2007.08159.

What is the role of the modulation factors? Is it just for inducing weight stability or are these factors necessary for continual learning? An ablation analysis would enable the reader to understand the role of the modulation factors? Generally modulation is considered necessary for continual learning and is helpful for selective attention in the network. Are the modulation factors performing a similar role?

6. The experimental setup in Section 4 is not clear. In Table 2, the accuracies for multi-head models looks inaccurate? Do you mean task incremental learning scenario for multihead or class Incremental learning? Class-IL based models are generally not categorized as multi-headed models? Please clarify the description and check the correctness in the accuracies for compared works. Additionally, newer works which demonstrate superior performance need to be compared. You can find some of the works in
    * Mai, Z., Li, R., Jeong, J., Quispe, D., Kim, H., & Sanner, S. (2022). Online continual learning in image classification: An empirical survey. Neurocomputing, 469, 28-51.


**Limitations:**

limitations have not been discussed in detail but potential negative societal impacts are briefly touched upon

**Strengths And Weaknesses:**

**Strengths**

* The back propagation-alternative approaches are closer to the biological neural networks and provide an avenue for efficient learning
* The treatment of local knowledge propagation among the hidden layers is novel.

**Weaknesses**
* The writing/presentation of ideas need to be improved for clarity
* The experimental setup is not clear/reproducible
* Comparison with other local learning-based continual learning approaches is missing

---

> ### Author Response · Authors · 2022-08-02
> **Response to Reviewer 36Ec (Part 1)**
>
> We would like to thank Reviewer 36Ec for their comments and questions.
>
> **Question 1:**
>
> Each latent state z^l (z^1, z^2,...,z^L) are initialized to vectors of zeros (see Line 4 of Algorithm 1). They become non-zero after K iterations of message passing / processing of the clamped input stimuli, i.e., the bottom layer of the S-NCN is clamped to z^0_x = x (and z^0_y = y during training). In the first iteration k = 1, the prediction vectors z^l_\mu would also be zero but the error neurons e^0_x and e^0_y would not be. Thus, lines 16-21 in Algorithm 1 would result in a non-zero z^1. After k = 2, z^2 would be non-zero, and so on and so forth (at least K = L iterations would be needed to obtain all non-zero latent states, e.g., K >= 3 for the models in this paper).
>
>
> **Question 2:**
>
> We agree that the text in our paper classifying our approach as “Domain-IL” was not accurate upon re-examining [1]. The S-NCN would actually be, according to [1] (Table 1, third line, which states “Class-IL | Solve tasks so far and infer task-ID”), classified as Class-IL given that it never uses task IDs/descriptors. Even though we mention task IDs in the problem formulation and use them for the many baselines we compare to, the S-NCN never uses them – the basal ganglia model is responsible for learning task contexts and mapping data to the right task, at both online training and test time. The original confusion on our end was the fact that the “class incremental” case falls under Class-IL and we did not learn one class at time (we learn 2 or more, depending on the benchmark). We emphasize that the S-NCN is not Task IL (which is arguably the simpler formulation of the continual learning problem).
> We will modify the paper to include the above correction/revision.
>
> [1] Van de Ven, Gido M., and Andreas S. Tolias. "Three scenarios for continual learning." (2019).
>
> **Question 3:**
>
> Yes, there is an overhead for the task-context memory – one new task context vector would be generated/grown as each new (disjoint) task is encountered. The reason we mention efficiency in our paper is that our S-NCN does not require growing out the parameters of the generative cortex/model itself (which, if we did so as in prior related work, many new parameters would need to be created per task as in Progressive Networks [1] or dynamically-expanding networks [2]). We reason that the inclusion of relatively small, additional context vectors is far more desirable and only requires growing out needed parameters at a rate far less than methods such as [1,2] (as mentioned in lines 103-104 in the original paper). Furthermore, the S-NCN does not require schemes for storing/organizing physical raw data as in the case of most replay methods. We remark that the S-NCN’s generative cortex could be adapted to induce its own form of layer-wise replay, similar to [3], but this has been left for future work (see “Discussion: On the Limitations of Sequential Neural Coding” in the original Appendix/Supplementary Material for this discussion).
> We will modify the paper/appendix to include the points raised above.
>
> [1] Rusu, Andrei A., et al. "Progressive neural networks."  (2016).
> [2] Yoon, Jaehong, et al. "Lifelong learning with dynamically expandable networks." (2017).
> [3] van de Ven, Gido M., Hava T. Siegelmann, and Andreas S. Tolias. "Brain-inspired replay for continual learning with artificial neural networks." (2020).
>
> **Question 4:**
>
> In contrast to surrogate gradient-based approaches, the S-NCN works without resorting to predicting gradients (as in decoupled neural interfaces [1] or DNIs) and directly resolves both the forward locking and update-locking problems. In effect, each layer-wise prediction can be made independently of the others (unlike typical forward passes in modern-day deep networks) and the synaptic updates for each layer (both forward and error synapses) can be made without others having been computed/completed. This opens the door to potential parallel asynchronous implementations of the S-NCN that could drastically speed up computation further. In contrast to DNIs, the S-NCN’s generative cortex does not require training gradient predictors (which typically require access to true gradients provided by backprop in order to train them properly) and, without incurring synthetic approximation errors (as in a fully-unlocked network using DNI, where now even the layer activities require additional modules to be trained to predict actual layer-wise activities) furthermore, resolves both update and forward locking . Crucially, the S-NCN’s updates are biologically plausible – it only requires simple (multi-term) Hebbian updates for the generative cortex and competitive Hebbian updates for the basal ganglia.
> We will modify the appendix to include the above question/answer.
>
> [1] Jaderberg, Max, et al. "Decoupled neural interfaces using synthetic gradients." (2017).

---

> > ### Author Response · Authors · 2022-08-02
> > **Response to Reviewer 36Ec (Part 2)**
> >
> > **Question 5:**
> >
> > The modulation factors are largely used, in this study, for improving weight/learning stability and, notably, allowed us to train the generative cortex without post-update weight normalization (which we found was not needed as it is sparse coding-like models – such synaptic normalization, i.e., normalizing synaptic matrix columns by their L2 norms, is harder to justify biologically in the brain whereas evidence has been provided for synaptic scaling [1, 2, 3], as mentioned in the appendix of the original submitted paper, i.e., Section "On Weight Update Modulation", which is what our modulation factors do). From a practical standpoint, this meant we did not have to craft an intricate synaptic normalization scheme (one that was aware of the task being operated on) and the synaptic scaling was found to be a bit faster to compute (per batch compared to normalization).
> > However, this question does spark an interesting future direction where the modulation factors could be driven by/conditioned on the FNBG circuit we proposed in this work (or another task-sensitive neural circuit) which would reduce forgetting during the synaptic update step.
> > We will include the above discussion in the Appendix as well as the references listed above/provided by Reviewer 36Ec in the appendix/paper.
> >
> > [1] G. G. Turrigiano, “The self-tuning neuron: synaptic scaling of excitatory synapses,” (2008).
> > [2] K. Ibata, Q. Sun, and G. G. Turrigiano, “Rapid synaptic scaling induced by changes in postsynaptic firing,” (2008).
> > [3] T. C. Moulin, et al., “The synaptic scaling literature: A ¨systematic review of methodologies and quality of reporting,” (2020).
> >
> > **Question 6:**
> >
> > There is actually a typo in the caption for Table 2. All models below the dashed line are single-head models while the ones above it are multi-head ones. We have fixed this error now in the revised/updated paper (this specific update has now been made to the current OpenReview draft).
> >
> > Furthermore, we have now, in response to Reviewer 36Ec, included three new modern baseline results from (Mai et al., 2022) as per Reviewer 36Ec’s request/suggestion, i.e., ER, A-GEM, and GDumb. Please see the revised/updated appendix for these additional modern baseline results (this specific update has now been made to the current OpenReview draft).
> > Note: for all of the baselines in this work, we performed a grid search over the learning rate, batch size, optimizer choice,  and hidden units to select optimal hyper-parameters by observing performance on the task validation set(s).
> >
> > **"limitations have not been discussed in detail but potential negative societal impacts are briefly touched upon"**
> >
> > Please see our paper’s Appendix/Supplementary Material for a detailed discussion on limitations, i.e., section titled “Discussion: On the Limitations of Sequential Neural Coding”.

---

> > ### Comment · Reviewer_36Ec · 2022-08-09
> > **Experimental settings not clear**
> >
> > The task-IL is the multi-head scenario where the task labels are used for training and testing scenarios, while the class-IL is the single head scenario, which task (with multiple classes) label  used only for training. It is mentioned in the response that experiments are Class-IL but I notice both multi-head and single head runs which is confusing.
> >
> > Also, what is the motivation for using 40 epochs for simple data like MNIST?
> >
> > Finally, most of the referred review papers and contemporary Continual Learning approaches use more complex data with a larger number of classes per task. How do the proposed approach work in that scenario. Does it scale to the larger tasks/sequences. Which in my opinion is important for the practicality of the continual learning approach.

---

> > > ### Author Response · Authors · 2022-08-10
> > > **Response to Reviewer 36Ec Response: Clarifying Settings Further**
> > >
> > > ***The task-IL is the multi-head scenario...It is mentioned in the response that experiments are Class-IL but I notice both multi-head and single head runs which is confusing.***
> > >
> > > To be clear, we were meaning that the proposed S-NCN is itself Class-IL - it is a single-head system that does not use task labels at all.
> > > However, all of the benchmarks contain task labels that can be used if desired (Split MNIST, for example, naturally contains task identifiers) -- this allows comparison to be made between single and multi-head models within the same setting. The multi-head/single-head models that required task labels (as there are single/multi-head models that use task labels) were allowed access to the task labels while the task-free models did not use them. Furthermore, we wanted to demonstrate that the S-NCN is competitive with methods (multi-head) that solve an easier version of the continual learning problem (hence we included them in the same table).
> > >
> > > ***...motivation for using 40 epochs for simple data like MNIST?***
> > >
> > > EWC and SI: Regularization-based models perform more stably, primarily when EWC is used with a single head. We observed on average 34 epochs are sufficient for SI and EWC to get the stable performance or minimal forgetting on prior tasks. Similar experiments were performed by the GEM paper where they show on Rotation MNIST ( 1 vs. 5 epochs) that EWC performance improves as the number of epochs increases. Third, performance on the current task is compromised when selecting a value for weight regularization factors or lamda for the prior task in EWC that is responsible for lower forgetting. Hence, we trained these systems for longer durations to achieve better performance on current tasks with lower forgetting. Thus we choose 40 for all our experiments; it should be noted approaches such as GEM, A-GEM, mnemonics, LUCIR, and other related methods can get decent performance within five epochs (while S-NCN only requires one).
> > >
> > > ***Finally, most...classes per task. How do the proposed approach work in that scenario. Does it scale to the larger tasks/sequences. Which in my opinion is important for the practicality of the continual learning approach.***
> > >
> > > In the context of the larger benchmarks, which generally focus on more complex images, one would need to utilize convolution to obtain acceptable accuracy/performance as often the larger task benchmarks involve using Cifar100, SVNH, and ImageNet (and beyond).
> > > This would be beyond the scope of this particular paper to develop a generalization to convolution (that is the subject of a concentrated follow-up effort) - although note this can be done (with local update rules for the kernels/filters) which would aid the S-NCN system to scale properly to very high-dimensional, natural image spaces (we can include a appendix discussion of this as this points to a nice bit of future work).
> > > However, the point/goal of this paper was not to engage in a complete performance contest but rather to demonstrate and provide empirical evidence that a neurobiologically-plausible complementary system (based on the interactions basal ganglia with predictive processing-based cortical circuits) is competitive on several benchmarks (as well as on a harder custom benchmark that tests a varying the number of classes incrementally observed, which many methods do not handle well) and exhibits minimal forgetting compared to a large set of baselines, including three new ones we added to address Reviewer 36Ec's earlier request. This is quite promising and important to encourage the community to examine the benefits that other less-explored elements of backprop-free, brain-inspired neural computation might bring to the continual machine learning scenario. Furthermore, we want to emphasize that each layer in the S-NCN generative circuit can work in parallel, offering a massive computational advantage, and the full complementary system operates purely online (in contrast to many modern-day methods that require multiple passes over each dataset).
> > >
> > > With respect to scaling to larger/longer task sequences, our method would scale reasonably well -- the only key requirement is that memory is available to grow out a task context memory (to allow the basal ganglia the extra parameters to manipulate as new disjoint tasks are encountered/detected) for each encountered disjoint task (for strongly similar or subsets of a similar task, the basal ganglia would just reuse previously created contexts as it would map similar tasks to each other) and that the generative (cortical) circuit is of high enough capacity (future work will investigate these capacity needs -- given that the brain contains billions neurons/parameters, one would ultimately desire a very high capacity generative circuit but one could take a more conservative neurogenesis approach and grow only as needed when a completely unrelated/disjoint task is encountered -- we will include this in the discussion in the Appendix).

---

> > > > ### Author Response · Authors · 2022-08-10
> > > > **Response to Reviewer 36Ec Response: Clarifying Settings Further (Addendum Note)**
> > > >
> > > > Note that we will also incorporate the above additional clarifications/question response (to Reviewer 36Ec) into the main paper/appendix.

---

### Author Response · Authors · 2022-08-03
**General Comment/Note for all Reviewers: 36Ec, WKzA, and qc3j**

We would like to thank all three reviewers for their comments and interesting questions. We have addressed all the typos and errors pointed out by all three reviewers and have modified our manuscript to include the requested additional results and content (we make an explicit note in each specific response to each reviewer of each point we will be sure to integrate/include).

Please find in the updated revised draft of both the ***main paper*** and the ***appendix/supplementary material*** the inclusion of what was requested/discussed by each reviewer. ***In both the main paper and appendix/supplementary material we have highlighted every portion of text in blue to make concrete and clear what we have changed/adjusted for the convenience of the reviewers*** and in order to aid them in examining the modified/added portions to see what was included.

We look forward to each one of your responses!

***Important Note:*** In addition to integrating much of the discussions and answers related to reviewer-specific questions in the appendix (note that we currently inserted discussion blocks/integrated in the appendix in order to respect the NeurIPS 9-page constraint -- we will move these bigger portions into the main paper upon moving on to the next phase), we have also included requested additional baseline results (****greedy sampler and dumb learner**** - GDumb, ****experience replay**** - ER, and ****average gradient episodic memory**** - A-GEM), ****task-accuracy plots**** for Split-MNIST/Fashion MNIST/NotMNIST for the S-NCN (at the very end of the appendix), and a ****derivation**** of the state and weight update dynamics from Equation 6 (this derivation can be found in the Appendix as well).

---

### Meta-Review · Area_Chair_jPzV · 2022-08-26

**Recommendation:** Accept
**Confidence:** Certain

**Metareview:**

This paper provides a biologically-inspired method based on predictive coding to address the dangers of catastrophic forgetting in continual learning, while encouraging models to leverage similar data from their past when learning from present information. The reviewers agreed the paper was interesting and worth publishing, although there was a spread in their enthusiasm. That said, the consensus is clear enough that I am happy to recommend acceptance.

**Award:**

No

---

### Decision · Program_Chairs · 2022-09-14

Accept